# Pushing the Boundaries of Natural Reasoning: Interleaved Bonus from Formal-Logic Verification in Language Models

**Chuxue Cao** [* 1 2]   **Jinluan Yang** [* 3]   **Haoran Li** [1]   **Kunhao Pan** [1]   **Zijian Zhao** [1]   **Zhengyu Chen** [3]
**Yuchen Tian** [4]   **Lijun Wu** [2]   **Conghui He** [2]   **Yike Guo** [1]   **Sirui Han** [1]

## Abstract

Large Language Models (LLMs) show remarkable capabilities, yet their stochastic next-token prediction creates logical inconsistencies and reward hacking that formal symbolic systems avoid. To bridge this gap, we introduce a formal logic verification-guided framework that dynamically interleaves formal symbolic verification with the natural language generation process, providing real-time feedback to detect and rectify errors as they occur. Distinguished from previous neuro-symbolic methods limited by passive post-hoc validation, our approach actively penalizes intermediate fallacies during the reasoning chain. We operationalize this framework via a novel two-stage training pipeline that synergizes formal logic verification-guided supervised fine-tuning and policy optimization. Extensive evaluation on six benchmarks spanning mathematical, logical, and general reasoning demonstrates that our 7B and 14B models outperform state-of-the-art baselines by average margins of 10.4% and 14.2%, respectively. These results validate that formal verification can serve as a scalable mechanism to significantly push the performance boundaries of advanced LLM reasoning. Our models and datasets are available at https://huggingface.co/collections/chuxuecao/flv.

## 1. Introduction

Large Language Models (LLMs) demonstrate impressive proficiency in mathematical and logical reasoning (Ahn et al., 2024; Ji et al., 2025; Liu et al., 2025b; Chen et al.,

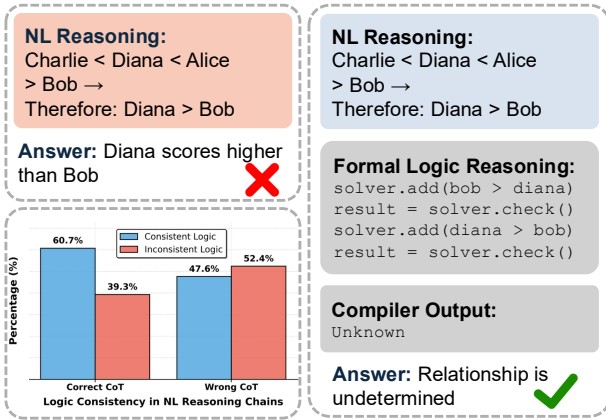

*Figure 1.* Comparison between natural language reasoning and formal logic verification-guided reasoning. Formal verification detects logical errors in natural language reasoning and provides corrected reasoning paths. "NL" means Natural Language.

2025a), yet their probabilistic decoding process lacks inherent mechanisms to ensure consistency (Sheng et al., 2025b; Baker et al., 2025). This tension creates significant risks, including hallucinations (Li & Ng, 2025; Sheng et al., 2025b), safety vulnerabilities (Zhou et al., 2025; Cao et al., 2025b), and reward hacking (Chen et al., 2025b; Baker et al., 2025). Although recent efforts have employed model-based verifiers to offer denser feedback than sparse ground-truth labels (Ma et al., 2025; Liu et al., 2025c; Guo et al., 2025b; Shi et al., 2025), they often overlook intermediate reasoning steps. To enforce more rigorous supervision, subsequent research has incorporated formal tools like theorem provers and code interpreters (Ospanov et al., 2025; Kamoi et al., 2025; Liu et al., 2025a) to address this drawback. However, existing formal approaches face critical limitations: they are often restricted to specific domains (e.g., Mathematics) (Ospanov et al., 2025; Liu et al., 2025a), rely on uncertain autoformalization (Ospanov et al., 2025; Feng et al., 2025b), or utilize post-hoc verification that cannot actively prevent error propagation (Kamoi et al., 2025; Feng et al., 2025b). This yields the primary question to be explored:

* Equal contribution. [1]Hong Kong University of Science and Technology [2]Shanghai Artificial Intelligence Laboratory [3]Zhejiang University [4]Hong Kong Baptist University. Correspondence to: Sirui Han <siruihan@ust.hk>, Yike Guo <yikeguo@ust.hk>.

*Proceedings of the 43rd International Conference on Machine Learning*, Seoul, South Korea. PMLR 306, 2026. Copyright 2026 by the author(s).

> *(Q) Can we utilize the formal verification to further enhance the LLM reasoning across diverse domains?*

To explore this question, we first quantified the logical consistency gap in current LLMs by conducting a formal verification analysis of generated reasoning chains. A critical finding emerges as shown in Figure 1: even chains that arrive at correct final answers suffer from substantial logical inconsistency, with 39.3% of steps formally disproved, a trend consistent with previous research (Sheng et al., 2025b; Leang et al., 2025). For chains leading to incorrect answers, this failure rate rises to 52.4%. The comparative example in Figure 1 illustrates this gap: natural language reasoning incorrectly concludes "Diana > Bob" from the given constraints, while formal verification identifies the incorrect conclusion and lead to an correct answer. This phenomenon reveals pervasive "reward hacking," where models exploit superficial patterns to reach correct labels without establishing sound logical foundations (Skalse et al., 2022). Ultimately, these results expose a fundamental limitation of natural language reasoning: without explicit verification mechanisms, models cannot guarantee reasoning validity or global logical coherence across multi-step inference.

To bridge this gap, we propose a novel framework that synergizes probabilistic natural language reasoning with formal symbolic verification. Distinguished from prior approaches relying on static filtering or narrow domains, our method dynamically interleaves formal verification into the generation process. By incorporating feedback from satisfiability results, counterexamples, and execution outputs, we extend the standard chain-of-thought paradigm to enable real-time error detection and rectification. To effectively operationalize this interleaved reasoning approach, we introduce a **formal logic verification-guided training framework** comprising two synergistic stages: (i) Supervised Fine-tuning (SFT), which employs a hierarchical data synthesis pipeline. Crucially, we mitigate the noise of raw autoformalization by enforcing execution-based validation, thereby ensuring high alignment between natural language and formal proofs; and (ii) Reinforcement Learning (RL), which utilizes Group Relative Policy Optimization (GRPO) (Shao et al., 2024) with a composite reward function to enforce structural integrity and penalize logical fallacies. Empirical evaluations across logical, mathematical, and general reasoning domains demonstrate that this framework substantially enhances reasoning accuracy, highlighting the potential of formal verification to push the performance boundaries of LLM reasoning. Our contributions can be concluded as follows:

- We propose the first framework that dynamically interleaves formal verification into LLM reasoning across diverse domains, utilizing the real-time feedback via symbolic interpreters to transcend the limitations of passive post-hoc filtering and domain-specific theorem proving.

- We introduce a two-stage training framework that combines formal verification-guided supervised fine-tuning with policy optimization, featuring a novel data synthesis pipeline with execution-based validation to enforce logical soundness and structural integrity.

- Extensive evaluations on six benchmarks demonstrate that our models break performance ceilings, surpassing SOTA by 10.4% (7B) and 14.2 % (14B). This validates the scalability and effectiveness of our proposed method.

## 2. Related Works

### 2.1. Large Language Models for Natural Reasoning

Supervised fine-tuning (SFT) on chain-of-thought examples (Wei et al., 2022) and step-by-step solutions (Cobbe et al., 2021) has been foundational for developing reasoning capabilities in LLMs, with recent efforts curating high-quality datasets across mathematics (LI et al., 2024), code (Xu et al., 2025), and science (Wang et al., 2022). However, SFT alone cannot effectively optimize for complex objectives beyond imitation and struggles with multi-step error correction (Lightman et al., 2023; Uesato et al., 2022; Zhou et al., 2026a). Recent RL advances using outcome-based optimization methods have achieved remarkable success in mathematical reasoning (Cobbe et al., 2021; Zeng et al., 2025), code generation (Le et al., 2022; Feng et al., 2025a), and general-domain reasoning (Ma et al., 2025; Chen et al., 2025c; Xiong et al., 2025; Zhu et al., 2026). However, optimizing solely for final answer correctness creates perverse incentives where models learn correct conclusions through logically invalid pathways (Uesato et al., 2022), leading to reward hacking (Skalse et al., 2022) and brittleness under distribution shift (Hendrycks et al., 2021). To address these limitations, process-based rewards incorporate feedback from intermediate steps, providing dense supervision through human-annotated judgments (Uesato et al., 2022; Lightman et al., 2023; She et al., 2025; Khalifa et al., 2025; ?; Zhou et al., 2026b). However, the probabilistic nature of language model-based verifiers introduces errors and biases (Zheng et al., 2023), limiting their ability to detect subtle logical inconsistencies that emerge during multi-step reasoning.

### 2.2. Formal Reasoning and Verification

Recent work has integrated formal verification tools, including theorem provers (Yang et al., 2023; Cao et al., 2025a; Tian et al., 2025), code interpreters (Feng et al., 2025a), and symbolic solvers (Li et al., 2025a), to provide machine-checkable validation beyond the biases of LLM-as-a-judge approaches (Li & Ng, 2025; Uesato et al., 2022; Lightman et al., 2023). This direction is increasingly recognized as

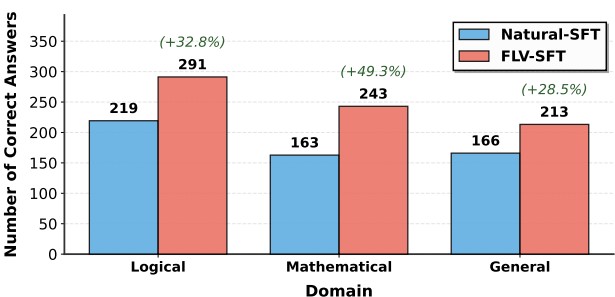

*Figure 2.* Number of correct answers using natural language reasoning versus formal logic verification. We randomly sampled 500 instances from each domain for this comparison.

critical for grounding generative models in verifiable systems (Ren et al., 2025; Wang et al., 2025; Hu et al., 2025). Existing approaches differ in how verification is applied. HERMES (Ospanov et al., 2025) interleaves informal reasoning with Lean-verified steps, ensuring real-time soundness but requiring mature formal libraries. Safe (Liu et al., 2025a) applies post-hoc verification to audit completed reasoning chains, though this passive mode cannot prevent error accumulation during generation. VeriCoT (Feng et al., 2025b) checks logical consistency on extracted first-order logic, while others train verifier models using formal tool signals (Kamoi et al., 2025). Tool-integrated methods (Xue et al., 2025; Zeng et al., 2025; Li et al., 2025b; Feng et al., 2025a) embed interpreter calls into generation for calculation or simulation, but often lack strict logical guarantees. These approaches face key limitations: specialization to mathematical theorem proving, treating verification as separate filtering without guiding generation, or relying on uncertain logic extraction and neural verifiers. In contrast, we propose verification-guided reasoning that extends formal verification to general logical domains and employs real-time feedback as dynamic, in-process guidance to steer reasoning trajectories and enable self-correction beyond specialized formal tasks.

# 3. Preliminaries

We first present empirical evidence illustrating the gap between probabilistic natural language reasoning and formal verification in LLM reasoning. We then formally define our reasoning paradigm and introduce the preliminaries on the symbolic verification methods utilized in our framework.

## 3.1. Natural vs. Formal Reasoning in LLMs

Our previous experiments demonstrate that LLMs lack mechanisms to ensure global logical consistency (Figure 1), motivating us to explore formal logic verification. Formal logic provides a rigorous framework where reasoning steps can be reliably validated using formal solvers. As shown in Figure 2, integrating formal logic verification with natural

language reasoning yields significant performance improvements across diverse domains. We compare two approaches: Natural-SFT, which relies solely on natural language reasoning, and FLV-SFT, which incorporates formal logic verification. Across 500 randomly sampled instances from each domain, FLV-SFT consistently outperforms Natural-SFT, achieving 291 vs. 219 correct answers in the Logical domain (+32.8%), 243 vs. 163 in the Mathematical domain (+49.3%), and 213 vs. 166 in the General domain (+28.5%). These substantial improvements across all three categories demonstrate that formal verification effectively addresses the consistency gaps inherent in purely neural approaches. These results underscore the significant potential of formal verification to bridge the reasoning gap and strongly motivate our approach of interleaving natural language reasoning with formal verification throughout the reasoning process.

## 3.2. Problem Formulation

Formally, let $\mathcal{D} = \{(x, y)\}$ be a dataset of reasoning tasks, where $x$ denotes the input context (e.g., problem description) and $y$ denotes the ground-truth answer.

**Standard CoT Paradigm.** In conventional Chain-of-Thought reasoning, an LLM $P_\theta$ generates a sequence of reasoning steps $z = (s_1, s_2, \ldots, s_n)$ in natural language, aiming to maximize:

$$P_\theta(y, z \mid x) = P_\theta(y \mid z, x) \cdot \prod_{i=1}^{n} P_\theta(s_i \mid s_{<i}, x) \quad (1)$$

However, this objective does not guarantee that $z$ is logically valid or formally verifiable.

**Our Paradigm: Formal Logic Verification-Guided Reasoning.** We propose augmenting the reasoning chain with formal verification at each step. Specifically, we define an extended reasoning chain $z' = (s_1, f_1, v_1, s_2, f_2, v_2, \ldots, s_n, f_n, v_n)$, where:

- $s_i$: Natural language reasoning step (as in standard CoT)
- $f_i$: Formal specification that encodes the logical correctness of $s_i$ (e.g., symbolic constraints, SAT clauses, SMT formulas, or executable code)
- $v_i$: Formal Logic Verification result returned by a formal verifier $\mathcal{V}$ when applied to $f_i$

During training, our objective is to maximize the likelihood of correct final answers:

$$\max_\theta \mathbb{E}(x, y) \sim \mathcal{D} \left[ \log P\theta(y, z' \mid x) \right] \quad (2)$$

During inference, the verification function $\mathcal{V}$ takes the formal reasoning as input and returns detailed feedback at each reasoning step. This feedback may include satisfiability results, counterexamples, proof traces, execution outputs, or

error messages, which guide the model to generate logically sound and verifiable subsequent reasoning steps.

# 4. Methodology

To address logical inconsistencies and hallucinations in pure natural language reasoning (Section 3), we propose integrating formal logic verification into the reasoning process. Our approach consists of two stages: (i) **Supervised Fine-tuning** to enable the model to generate interleaved natural language and formal proofs, and (ii) **Reinforcement Learning** to optimize the model using composite rewards that enforce logical soundness and correctness.

## 4.1. Formal Logic Verification-Guided SFT

The primary goal of the SFT stage is to align the model's output distribution with a structured reasoning format that supports self-verification. Since large-scale datasets containing interleaved reasoning and formal proofs are scarce, we employ a hierarchical formal proof data synthesis pipeline.

### 4.1.1. DATA SYNTHESIS PIPELINE

**CoT Generation.** Given a raw reasoning problem $q$, we first employ a strong teacher model to generate $K = 4$ candidate reasoning chains. A judge model evaluates the correctness of the final answers to calculate pass rates. We select a subset of verified chains that yield correct solutions for subsequent processing. Let $z$ denote a selected correct reasoning chain. To incorporate formal logic, we utilize an LLM to decompose $z$ into discrete logical modules $\{s_k\}_{k=1}^N$. For each module $s_k$, the LLM synthesizes a corresponding formal proof $f_k$ and predicts an expected execution output $v_k^{\exp}$. The prompt template is provided in Figure 13.

**Execution-based Validation and Correction.** To ensure the fidelity of synthesized formal proofs, we implement a rigorous validation mechanism. Each generated formal proof $f_k$ is executed in a sandbox to obtain the actual output $v_k^{\rm act}$. We then perform a three-stage validation:

*Stage 1: Exact Match.* If the actual output exactly matches the expected output ($v_k^{\rm act} = v_k^{\exp}$), the proof is accepted immediately and integrated into the training data.

*Stage 2: Semantic Equivalence Check.* In cases where $v_k^{\rm act} \neq v_k^{\exp}$, we employ a verification model to assess whether the discrepancy is semantically negligible (e.g., differences in capitalization, output ordering, or numerical precision). If the outputs are deemed equivalent under mathematical or logical semantics, we proceed to Stage 3.

*Stage 3: Proof Rewriting.* When minor inconsistencies are detected, we require the LLM to regenerate the natural language reasoning $s_k'$ conditioned on the actual execution result $v_k^{\rm act}$. This ensures that the natural language reasoning

module $s_k'$, the formal proof $f_k'$, and the execution output $v_k^{\rm act}$ maintain strict logical coherence. Proofs that fail both exact match and semantic equivalence checks are discarded. The resulting training instance is structured as follows:

$$
z_{\rm aug} = \bigoplus_{k=1}^N \Big( s_k \oplus \texttt{} f_k' \texttt{} \\
\oplus \texttt{<interpreter>} v_k^{\rm act} \texttt{</interpreter>} \Big)
\tag{3}
$$

where $f_k'$ denotes the validated formal proof and $v_k^{\rm act}$ is the verified execution output. This pipeline ensures that every training example $(s, f, v)$ exhibits perfect alignment between natural language hypotheses, formal logic reasoning, and execution feedback, thereby providing high-quality supervision for the model to learn reliable reasoning patterns. See Appendix B for dataset construction details.

### 4.1.2. OPTIMIZATION OBJECTIVE

Given the augmented training dataset $\mathcal{D}_{\rm SFT} = \{(q_i, z_{{\rm aug},i})\}_{i=1}^M$, we optimize the model parameters $\theta$ by maximizing the log-likelihood of generating structured reasoning sequences:

$$
\mathcal{L}_{\rm SFT}(\theta) = -\mathbb{E}_{(q,z_{\rm aug})\sim\mathcal{D}_{\rm SFT}} \left[\log P_\theta(z_{\rm aug} \mid q)\right]
\tag{4}
$$

This can be decomposed into the sequential generation of reasoning modules, formal proofs, and execution outputs:

$$
\mathcal{L}_{\rm SFT}(\theta) = -\mathbb{E}_{(q,z_{\rm aug})\sim\mathcal{D}_{\rm SFT}} \left[ \sum_{k=1}^N \Big( \log P_\theta(s_k \mid q, z_{<k}) \right. \\
\left. + \log P_\theta(f_k' \mid q, z_{<k}, s_k) + \log P_\theta(v_k^{\rm act} \mid q, z_{<k}, s_k, f_k') \Big) \right]
\tag{5}
$$

where $z_{<k}$ denotes all previous modules. We train using AdamW with cosine learning rate scheduling and gradient clipping. This stage enables the model to generate verifiable, formally grounded reasoning chains.

## 4.2. Formal Verification-Guided Policy Optimization

To further enhance the formal logic verification-guided reasoning capabilities of LLMs, we employ reinforcement learning. The core of this stage is a multi-dimensional reward function that provides fine-grained feedback on structure, semantics, and computational efficiency.

### 4.2.1. HIERARCHICAL REWARD DESIGN

To ensure both generation stability and reasoning rigor, we design a hierarchical reward function $R(y)$ that evaluates responses in a strictly prioritized order: *Format Integrity* $\succ$

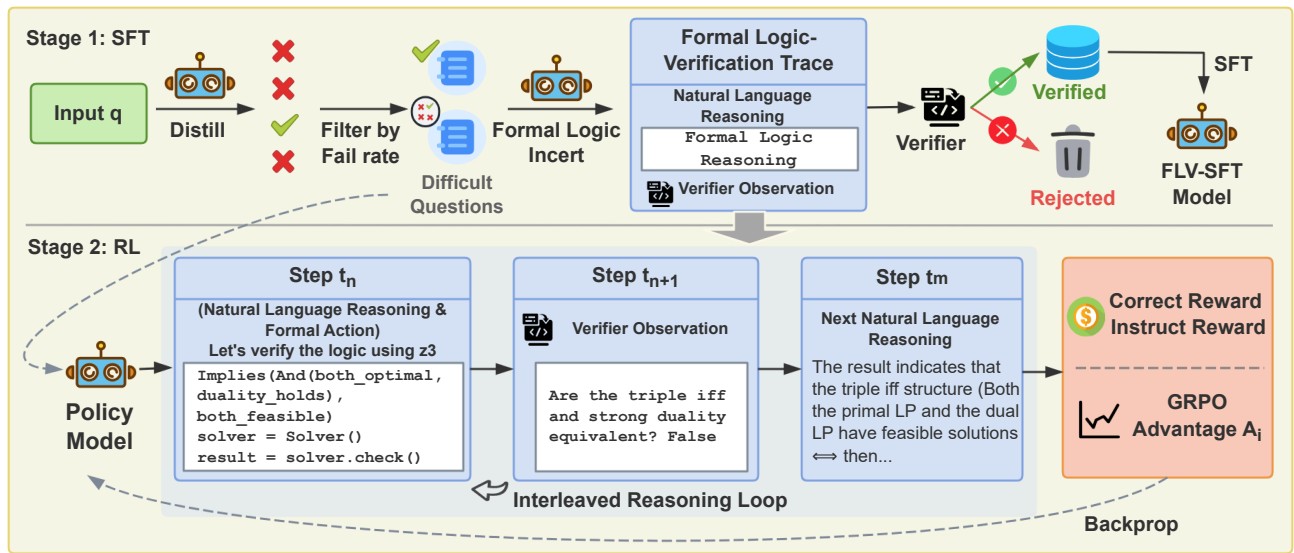

*Figure 3.* Overview of the formal logic verification-guided training framework. The framework operates in two stages: (1) **SFT:** A teacher model synthesizes formal logic verification traces, which are validated by a verifier. A subset of verified samples is used to fine-tune the model, while challenging samples are reserved for RL training. (2) **RL:** The policy model generates natural language reasoning followed by formal reasoning. A formal interpreter verifies the formal reasoning and provides feedback, enabling iterative refinement until the model produces a final answer or reaches the maximum number of interpreter calls. Rewards computed from verification outcomes are used to calculate advantages and update the policy model via reinforcement learning.

*Structural Compliance $\succ$ Logical Correctness*. The unified reward is formulated as:

$$R(y) = \begin{cases} R_{\text{fatal}} & y \in \mathbb{C}_{\text{fatal}} \quad \text{(L1: Fatal)} \\ R_{\text{invalid}} & y \in \mathbb{C}_{\text{invalid}} \quad \text{(L2: Format)} \\ R_{\text{total}}(y) & \text{otherwise} \quad \text{(L3: Valid)} \end{cases} \quad (6)$$

The total reward for valid responses is:

$$R_{\text{total}}(y) = R_{\text{struct}}(y) + R_{\text{logic}}(y) \quad (7)$$

**Level 1 & 2: Penalties for Malformed Generation.** We first filter out pathological behaviors to prevent reward hacking and infinite loops during training.

- **Fatal Errors ($\mathbb{C}_{\text{fatal}}$):** Responses with severe and unrecoverable execution failures (e.g., timeouts, repetition loops, excessive tool calls). We assign a harsh penalty $R_{\text{fatal}} = -\gamma_{\text{struct}} - W$ to strictly inhibit these states, where $W > 0$ is a correctness weight hyperparameter.

- **Format Violations ($\mathbb{C}_{\text{invalid}}$):** Responses that are technically executable but structurally flawed (e.g., missing termination tags, no extractable final answer, excessive verbosity). These incur a moderate penalty $R_{\text{invalid}} = -\beta_{\text{struct}} - W$, where $\gamma_{\text{struct}} > \beta_{\text{struct}} > 0$.

**Level 3: Incentives for Valid Reasoning.** For responses that pass the format checks ($y \notin \mathbb{C}_{\text{fatal}} \cup \mathbb{C}_{\text{invalid}}$), the reward is a composite of structural quality and logical correctness.

*(i) Structural Reward $R_{struct}(y)$:* Encourages concise and compliant tool usage.

$$R_{\text{struct}}(y) = \alpha - \lambda_{\text{tag}} \cdot N_{\text{undef}} - \lambda_{\text{call}} \cdot \max(N_{\text{call}} - N_{\text{max}}, 0) \quad (8)$$

Here, $\alpha$ is a base bonus, $N_{\text{undef}}$ tracks undefined tags, and the last term penalizes excessive tool invocations beyond a threshold $N_{\text{max}}$.

*(ii) Logical Correctness Reward $R_{logic}(y)$:* Evaluates the final answer $\hat{a}$ against the ground truth $a^*$.

$$R_{\text{logic}}(y) = \begin{cases} W - \lambda_{\text{len}} \cdot \Delta_{\text{len}}(\hat{a}, a^*) & \text{if } \hat{a} = a^* \\ -W & \text{if } \hat{a} \neq a^* \end{cases} \quad (9)$$

where $\Delta_{\text{len}}$ penalizes length discrepancies to discourage verbose reasoning and promote conciseness. Detailed hyperparameter settings are provided in Appendix C.

### 4.2.2. OPTIMIZATION OBJECTIVE

We optimize $\pi_\theta$ using GRPO. For each input $x \sim \mathcal{D}_{\text{difficult}}$, we sample $G$ outputs $y_1, \ldots, y_G$ and compute:

$$\mathcal{L}_{\text{GRPO}}(\theta) = \mathbb{E}_{x \sim \mathcal{D}_{\text{difficult}}} \left[ \frac{1}{G} \sum_{i=1}^{G} \sum_{t=1}^{|y_i|} \Big( \right.$$

$$\min \left( r_{i,t} \hat{A}_i, \text{clip}(r_{i,t}, 1 - \epsilon, 1 + \epsilon) \hat{A}_i \right) \quad (10)$$

$$\left. - \beta \log \frac{\pi_\theta(y_{i,t}|x, y_{i,<t})}{\pi_{\text{ref}}(y_{i,t}|x, y_{i,<t})} \Big) \right]$$

*Table 1.* Comparative evaluation between our proposed formal-language verification (FLV) methods (gray background) and other approaches. **Bold** values denote the best results. KOR-Bench and BBH contain multiple subfields and report macro-averaged scores. [†]CoRT-1.5B-RL is based on DeepSeek-R1-Distill-Qwen-1.5B.

| Model | Logical | | Mathemathcal | | General | | AVG |
|---|---|---|---|---|---|---|---|
| | KOR-Bench | BBH | MATH500 | AIME24 | GPQA-D | TheoremQA | |
| **Qwen2.5-7B** | | | | | | | |
| Base | 13.2 | 41.9 | 60.2 | 6.5 | 29.3 | 29.1 | 30.0 |
| Qwen2.5-7B-Instruct | 40.2 | 67.0 | 75.0 | 9.4 | 33.8 | 36.6 | 43.7 |
| SimpleRL-Zoo | 34.2 | 59.8 | 74.0 | 14.8 | 24.2 | 43.1 | 41.7 |
| General-Reasoner | 43.9 | 61.9 | 73.4 | 12.7 | **38.9** | 45.3 | 46.0 |
| RLPR | 42.2 | 66.2 | 77.2 | 14.2 | 37.9 | 44.3 | 47.0 |
| Synlogic | 48.1 | 66.5 | 74.6 | 15.4 | 27.8 | 39.2 | 45.3 |
| ZeroTIR | 30.0 | 40.0 | 62.4 | 28.5 | 28.8 | 36.4 | 37.7 |
| SimpleTIR | 37.0 | 62.0 | **82.6** | **41.0** | 22.7 | 51.1 | 49.4 |
| CoRT-1.5B-RL[†] | 28.0 | 41.0 | 76.8 | 25.2 | 8.6 | 46.5 | 37.7 |
| FLV-SFT (Ours) | 48.0 | 68.5 | 77.2 | 20.0 | 32.3 | 53.0 | 49.8 |
| FLV-RL (Ours) | **51.0** | **70.0** | 78.6 | 20.8 | 35.4 | **55.7** | **51.9** |
| **Qwen2.5-14B** | | | | | | | |
| Base | 37.4 | 52.0 | 65.4 | 3.6 | 32.8 | 33.0 | 37.4 |
| Qwen2.5-14B-Instruct | 51.5 | 72.9 | 77.4 | 12.2 | 41.4 | 41.9 | 49.6 |
| SimpleRL-Zoo | 37.2 | 72.7 | 77.2 | 12.9 | 39.4 | 48.9 | 48.1 |
| General-Reasoner | 41.3 | 71.5 | 78.6 | 17.5 | **43.4** | 55.3 | 51.3 |
| FLV-SFT (Ours) | 54.0 | 77.5 | 79.8 | 21.9 | 40.4 | 60.6 | 55.7 |
| FLV-RL (Ours) | **57.0** | **78.0** | **81.4** | **30.2** | 41.4 | **63.5** | **58.6** |

where $r_{i,t} = \pi_\theta(y_{i,t}|x, y_{i,<t})/\pi_{\text{old}}(y_{i,t}|x, y_{i,<t})$. The advantage $\hat{A}_i$ is group-normalized on $R(y_i)$, stabilizing training by emphasizing relative quality over absolute reward.

## 5. Experiment

### 5.1. Experimental Setup

**Models.** We utilize the Qwen2.5-7B and Qwen2.5-14B (Qwen-Team, 2025) as our backbone architectures. These models serve as the initialization point for both the SFT and subsequent Policy Optimization stages.

**Training Data.** Our training corpus is constructed using three datasets: WebInstruct-Verified (Ma et al., 2025), K&K (Xie et al., 2024), and NuminaMath-TIR (LI et al., 2024). These sources provide a collection of diverse, verifiable reasoning tasks across multiple domains. We employ DeepSeek-R1 (Guo et al., 2025a) for data distillation and difficulty assessment via pass-rate. We utilize GPT-4o (Hurst et al., 2024) as a judge for answer correctness and Claude-Sonnet-4.5 (Anthropic, 2024) to synthesize the interleaved formal logic steps as detailed in Section 4. The RL data is selected based on the pass rate of answers generated by the teacher model DeepSeek-R1, retaining only questions with a pass rate below 50%. The categorical distribution of our curated training data is illustrated in Figure 7.

**Evaluation.** We conduct a comprehensive evaluation across three distinct reasoning domains to assess models:

- **Logical Reasoning:** We employ KOR (Ma et al., 2024) to evaluate knowledge-grounded logical reasoning

across diverse domains and BBH (Suzgun et al., 2023) for challenging tasks requiring multi-step deduction.

- **Mathematical Reasoning:** We evaluate performance on MATH-500 (Hendrycks et al., 2024) for competition-level mathematics problems and AIME 2024 for Olympiad-level mathematical reasoning challenges.

- **General Reasoning:** We utilize GPQA-Diamond (Rein et al., 2024) for graduate-level reasoning in subdomains including physics, chemistry, and biology. Additionally, we use TheoremQA (Chen et al., 2023) to assess graduate-level theorem application across mathematics, physics, Electrical Engineering & Computer Science (EE&CS), and Finance, testing the model's ability to correctly apply and reason with formal theorems.

All evaluations use OpenCompass (Contributors, 2023) with greedy decoding, except AIME24 which reports avg@16 from sampling runs following RLPR (Yu et al., 2025).

**Baselines.** To validate the effectiveness of our framework, we compare our approach against five categories of models: (i) **Base Models:** Qwen2.5-7B/14B (Qwen-Team, 2025), establishing the baseline performance to measure the net gain of our training methodology. (ii) **Simple-RL-Zoo** (Zeng et al., 2025), a comprehensive collection of mathematics-focused RL models. (iii) **General-Reasoner** (Ma et al., 2025), a suite of general-domain RL models trained using a model-based verifier. (iv) **RLPR-7B** (Yu et al., 2025), a general-domain RL model trained via a simplified verifier-free framework. (v) **SynLogic-7B** (Liu et al., 2025b), a spe-

cialized model trained to enhance the logical reasoning capabilities of LLMs. (vi) **ZeroTIR** (Mai et al., 2025), a tool-integrated reasoning model specifically designed to execute Python code for solving mathematical problems. (vii) **SimpleTIR** (Xue et al., 2025), a multi-turn tool-integrated reasoning model for mathematical reasoning problems. (viii) **CoRT-1.5B-RL** (Li et al., 2025a), a tool-integrated reasoning model based on DeepSeek-R1-Distill-Qwen-1.5B that interleaves code execution within the thinking process.

**Implementation Details.** We implement our RL training using the verl framework (Sheng et al., 2025a), following ToRL (Li et al., 2025b). SFT Stage: We use a learning rate of $1 \times 10^{-5}$ with cosine scheduling and a global batch size of 32. The model is trained for 3 epochs. RL Stage: We employ a learning rate of $5 \times 10^{-7}$. We generate 8 rollouts per prompt with a temperature of 1.0. To prevent policy divergence, we set the KL coefficient to 0.05 and the clip ratio to 0.3. The training utilizes a batch size of 1024 and a context length of 16,384 tokens. Training proceeds for 120 steps on a cluster of 16 NVIDIA H800 GPUs. To manage computational overhead, we limit the formal verification process to a maximum of 4 iterative rounds.

### 5.2. Main Results

Table 1 presents the comprehensive evaluation of our Formal Logic Verification (FLV) approach against standard baselines across Qwen2.5-7B and 14B scales.

**Formal logic verification-guided methods outperform traditional natural language-based methods.** As shown in the results, our proposed FLV framework demonstrates superior performance compared to standard natural language reasoning approaches. Notably, even our supervised fine-tuning stage (**FLV-SFT**) surpasses all comparative RL baselines on the 7B scale. On Qwen2.5-7B, FLV-SFT achieves an average score of 49.8, outperforming the strongest natural language baseline (RLPR, 47.0) by 2.8 points. This suggests that integrating formal logic verification during the SFT phase provides a more robust reasoning foundation than standard RL training on natural language chains alone. Furthermore, the application of Group Relative Policy Optimization (**FLV-RL**) yields consistent improvements over the SFT stage. For the 14B model, FLV-RL improves upon FLV-SFT by increasing the average score from 55.7 to 58.6, with significant gains in hard mathematical tasks like AIME 2024 (+8.3%) and general theorem application in TheoremQA (+2.9%). This confirms that our verifier-guided RL effectively refines the policy beyond the supervised baseline.

**Formal logic verification unlocks model reasoning potential, achieving SOTA with limited data.** Despite utilizing a concise training set (approx. 17k samples total), our approach establishes new state-of-the-art performance among models of similar size, significantly outperforming

baselines that typically rely on larger-scale data consumption. (i) On the challenging AIME 2024 benchmark, our FLV-RL-14B model achieves 30.2%, nearly doubling the performance of the General-Reasoner baseline (17.5%) and far exceeding the Base model (3.6%). Similarly, on MATH-500, we achieve 81.4%, surpassing all baselines. (ii) We observe dominant performance on TheoremQA (63.5% on 14B), outperforming the nearest competitor by over 8 points. In logical reasoning (KOR-Bench), our method achieves a 15.7% improvement over the General-Reasoner on the 14B scale (57.0 vs 41.3). While FLV shows a slight weakness on GPQA-Diamond (likely due to benchmark reliability issues discussed in Appendix J), our method consistently excels in tasks requiring rigorous multi-step deduction and symbolic manipulation, validating the hypothesis that formal verification serves as a catalyst for deep reasoning capabilities.

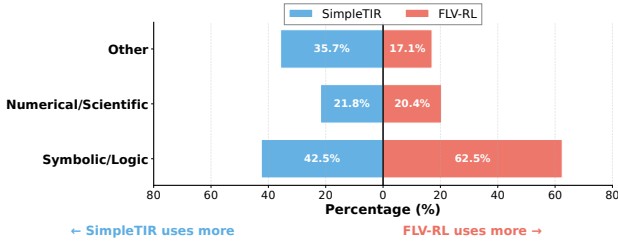

*Figure 4.* Python packages type distribution invoked by SimpleTIR (blue) vs. FLV-RL (red) across three domains.

**Formal verification instills a shift from calculation to symbolic reasoning, enabling superior generalization.** While tool-integrated baselines like SimpleTIR primarily utilize tools as "solvers" for direct computation (achieving 41.0 on AIME24), this paradigm struggles with tasks requiring rigorous logical deduction. In contrast, our FLV framework employs formal methods as a "verifier" to enforce logical consistency. This approach yields dominant performance on logic-heavy benchmarks such as KOR-Bench (51.0 vs. 37.0 for SimpleTIR) and GPQA-Diamond (35.4 vs. 28.8 for ZeroTIR). To understand the mechanism behind this reliability, we analyze the distribution of invoked Python packages in Figure 4. The data reveals a distinct behavioral shift: whereas SimpleTIR relies significantly on generic utility packages (Other), FLV-RL demonstrates a massive surge in the usage of Symbolic/Logic libraries. These formal tools constitute 62.5% of FLV-RL's calls—a 20-point increase over SimpleTIR. Meanwhile, the usage of Numerical/Scientific libraries remains stable ($\sim$21%), indicating that our method's gains are driven specifically by the adoption of symbolic logic engines to verify reasoning processes, rather than merely computing numerical answers. See Appendix 9 for the package categorization principles.

**Efficiency Analysis.** We analyze the computational cost of our approach by comparing the token length distributions of the natural language baseline (General-Reasoner), tool-

*Table 2.* We compare the performance of our proposed method (FLV) against natural language baselines across two training stages: SFT and GRPO. **Natural-SFT/GRPO** denotes models trained on the same data but without formal logic verification components. **FLV-SFT/GRPO** denotes our method incorporating formal logic modules and execution feedback.

| Model | Logical | | Mathematical | | General | | AVG |
|---|---|---|---|---|---|---|---|
| | KOR-Bench | BBH | MATH500 | AIME24 | GPQA-D | TheoremQA | |
| Base | 13.2 | 41.9 | 60.2 | 6.5 | 29.3 | 29.1 | 30.0 |
| Natural-SFT | 30.4 | 55.9 | 56.6 | 8.5 | 27.3 | 39.1 | 36.3 |
| FLV-SFT (Ours) | 48.0 | 68.5 | 77.2 | 20.0 | 32.3 | 53.0 | 49.8 |
| Natural-RL | 35.7 | 55.4 | 54.4 | 4.8 | 30.3 | 41.2 | 37.0 |
| FLV-ZeroRL | 48.0 | 55.0 | 71.0 | 10.2 | 22.7 | 49.1 | 42.7 |
| FLV-RL (Ours) | **51.0** | **70.0** | **78.6** | **20.8** | **35.4** | **55.7** | **51.9** |

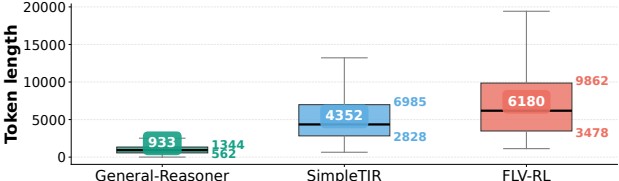

*Figure 5.* Token length distribution comparison across General-Reasoner, SimpleTIR, and FLV-RL. The box plots illustrate the median token usage (center line) and interquartile ranges.

integrated reasoning (SimpleTIR), and our FLV-RL method (Figure 5). While FLV-RL incurs a moderate computational overhead, we argue that this cost is justified by the substantial performance gains observed across diverse domains. The increased token consumption represents a necessary trade-off for achieving breakthrough generalization and ensuring logical soundness in high-stakes reasoning tasks.

### 5.3. Ablation Studies

To evaluate the individual contributions of our proposed components, we conducted an ablation study examining two critical dimensions: the impact of FLV versus pure natural language reasoning, and the effectiveness of different training paradigms. Results are presented in Table 2.

**Impact of formal logic verification.** Comparing FLV-based models against natural language baselines trained on identical data reveals substantial improvements. FLV-SFT achieves 49.8% average accuracy versus 36.5% for Natural-SFT, with particularly strong gains on logic-intensive tasks (KOR-Bench: +16.2 points, TheoremQA: +13.9 points). This demonstrates that formal proofs and execution validation fundamentally improve reasoning by grounding outputs in verifiable logic rather than probabilistic patterns.

**Impact of multi-stage training** We can observe that supervised fine-tuning establishes strong foundations, improving from 30.0% (Base) to 49.8% (FLV-SFT). Policy optimization yields further substantial gains to 51.9% (FLV-RL). Notably, natural language baselines barely improve with RL (37.0% vs 36.5%), while FLV-RL substantially outperforms FLV-SFT, indicating formal verification provides more sta-

ble and reliable reward signals for policy optimization.

**Necessity of SFT cold start.** To confirm that the formal verification capability is genuinely learned rather than an elicitation of pre-existing coding skills, we run a FLV-ZeroRL baseline that applies RL directly from the base model without any SFT initialization. This ablation yields only marginal gains over the base model (Base: 30.3 → FLV-ZeroRL: 42.7 → FLV-RL: 51.9), confirming two things: (i) the SFT cold start is essential for teaching the model the "Formal Tool-as-Verifier" paradigm, and (ii) the integration of formal verification and calculation is a newly learned capability rather than a simple elicitation of inherent coding skills.

### 5.4. Verification Paradigm: Balancing Formalism and Computational Fluency

Our initial data construction enforced explicit verification outputs (e.g., `proved`/`disproved`) after each logical module. However, this rigid format introduced two critical issues: (i) formal language redundancy, and (ii) suppression of direct calculation. When computational steps were needed, models would bypass direct arithmetic in favor of indirect validation via `z3-solver` (e.g., asserting `A + B == C is proved` rather than computing the sum), significantly degrading mathematical performance. To address this, we adopted a **flexible verification** strategy that decouples calculation from validation: (i) *Calculation as inference*: models invoke numerical tools directly without mandatory verification keywords; (ii) *Logic as validation*: formal verification serves as post-hoc validation for logical steps rather than a per-step constraint. This decoupling requires no changes to the inference pipeline—the underlying `natural reasoning` → `code` → `tool execution` loop remains identical; the shift is reflected solely in what the model generates within this loop. Figure 6 compares performance across logic, general, and math subsets under both paradigms. The flexible approach substantially improves math scores while preserving logical reasoning capability. Representative cases are detailed in Appendix E.

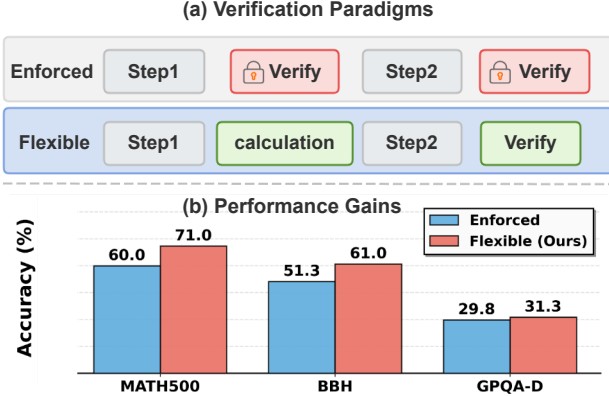

*Figure 6.* Enforced vs. Flexible Verification Paradigms. (a) Enforced verification imposes rigid checkpoints throughout the reasoning process, while flexible verification enables adaptive utilization of logic verification. (b) Performance gains after switching to flexible reasoning across three representative benchmarks.

## 6. Conclusion

In this work, we addressed the fundamental tension between probabilistic language generation and logical consistency in LLM reasoning by introducing a framework that dynamically integrates formal logic verification into the reasoning process. Through our two-stage training methodology combining FLV-SFT's rigorous data synthesis pipeline with formal logic verification-guided policy optimization, we demonstrated that real-time symbolic feedback can effectively mitigate logical fallacies that plague standard Chain-of-Thought approaches. Empirical evaluation across six diverse benchmarks validates our approach, with our 7B and 14B models achieving average improvements of 10.4% and 14.2% respectively over SOTA baselines, while providing interpretable step-level correctness guarantees. Beyond performance gains, our framework establishes a principled foundation for trustworthy reasoning systems by bridging neural fluency with symbolic rigor, thereby enabling more robust logical inference. This opens pathways toward more reliable AI that scales effectively to complex real-world problems across domains requiring strict logical soundness.

## Acknowledgements

We thank Liang Chen for his assistance with the experiments. We also sincerely thank the anonymous reviewers for their valuable feedback, which greatly helped improve the quality of this paper. This work is funded in part by the HKUST Start-up Fund (R9911), Theme-based Research Scheme grant (T45-205/21-N), the InnoHK initiative of the Innovation and Technology Commission of the Hong Kong Special Administrative Region Government, and the research funding under HKUST-DXM AI for Finance Joint Laboratory (DXM25EG01).

## Impact Statement

This paper presents work whose goal is to advance the field of Machine Learning. There are many potential societal consequences of our work, none which we feel must be specifically highlighted here.

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

## Limitations

Despite significant improvements in logical reasoning capabilities, our framework faces two primary limitations. First, integrating real-time formal verification introduces computational overhead, approximately doubling RL training time relative to the tool-free General-Reasoner baseline. However, this cost is acceptable given the substantial performance gains (10.4%-14.2% improvement) and superior data efficiency—we achieve comparable results using only a fraction of the training data required by existing methods, such that reduced data collection costs offset the increased training time. Second, our data synthesis pipeline faces formalization challenges when translating natural language into verifiable formal representations. While conversion success rates are high in structured domains like mathematics and logic, ambiguous or commonsense-heavy descriptions may produce mapping errors that generate incorrect verification feedback, limiting generalizability to open-ended reasoning tasks and highlighting the need for more robust auto-formalization techniques.

## A. Reward Calculation Pseudocode

*Table 3.* Hierarchical reward function for formal logic verification-guided policy optimization

| **Algorithm: Hierarchical Reward Computation** |
| --- |
| **Input:** Output $y$, Ground truth answer $a^*$, Predicted answer $\hat{a}$ 
 **Output:** Total reward $R(y)$ |
| **Hyperparameters:** 
 $\quad W = 3$ (correctness weight) 
 $\quad \gamma_{\text{struct}} = 3.0,\ \beta_{\text{struct}} = 1.0$ (severity penalties) 
 $\quad \alpha = 1.0$ (base structural score) 
 $\quad \lambda_{\text{tag}} = 0.005,\ \tau_{\text{tag}} = 200$ (tag penalty coefficients) 
 $\quad \lambda_{\text{call}} = 0.5,\ N_{\text{max}} = 3$ (tool call limits) 
 $\quad \lambda_{\text{len}} = 0.04,\ \delta_{\text{max}} = 10$ (length penalty) |
| **Step 1: Check Fatal Errors** ($\mathbb{C}_{\text{fatal}}$) 
 $\quad$ **if** token-level repetition detected **or** 
 $\quad\quad$ execution timeout **or** 
 $\quad\quad$ tool calls $> 2 \times N_{\text{max}}$ **or** 
 $\quad\quad$ multiple termination tags **then** 
 $\quad\quad$ **return** $R(y) = -\gamma_{\text{struct}} - W = -8.0$ |
| **Step 2: Check Invalid Format** ($\mathbb{C}_{\text{invalid}} \setminus \mathbb{C}_{\text{fatal}}$) 
 $\quad$ **if** solution extraction fails **or** 
 $\quad\quad$ solution length $> 512$ tokens **or** 
 $\quad\quad$ missing closing tag **or** 
 $\quad\quad$ $N_{\text{max}} <$ tool calls $\leq 2 \times N_{\text{max}}$ **then** 
 $\quad\quad$ **return** $R(y) = -\beta_{\text{struct}} - W = -6.0$ |
| **Step 3: Compute Structural Reward** $R_{\text{struct}}(y)$ 
 $\quad N_{\text{undef}} =$ count of undefined tags 
 $\quad N_{\text{call}} =$ count of tool invocations 
 $\quad R_{\text{struct}}(y) = \alpha - \lambda_{\text{tag}} \cdot \min(N_{\text{undef}}, \tau_{\text{tag}})$ 
 $\quad\quad\quad\quad -\lambda_{\text{call}} \cdot \max(N_{\text{call}} - N_{\text{max}}, 0)$ |
| **Step 4: Compute Correctness Reward** $R_{\text{correct}}(y)$ 
 $\quad f_{\text{len}}(\hat{a}, a^*) = \min(\lvert \text{len}(\hat{a}) - \text{len}(a^*) \rvert, \delta_{\text{max}})$ 
 $\quad$ **if** $\hat{a}$ matches $a^*$ **then** 
 $\quad\quad R_{\text{correct}}(y) = W - \lambda_{\text{len}} \cdot f_{\text{len}}(\hat{a}, a^*)$ 
 $\quad$ **else** 
 $\quad\quad R_{\text{correct}}(y) = -W$ |
| **Step 5: Compute Total Reward** 
 $\quad R(y) = R_{\text{struct}}(y) + R_{\text{correct}}(y)$ |

Table 3 provides the complete algorithmic implementation of our multi-component reward function used in FLV-RL training. The pseudocode details the step-by-step computation of format rewards, correctness rewards, and formal verification rewards, including all constraint checks and penalty mechanisms described in Section 4.

The time complexity for calculating the reward $R(y)$ is dominated by the verification of structural constraints and semantic correctness. Let $L$ denote the length of the generated response $y$ in tokens. The initial screening for pathological states ($\mathbb{C}_{\text{fatal}}$) and invalid formats ($\mathbb{C}_{\text{invalid}}$) requires a linear scan of the output tokens to detect repetition loops, count tool invocations ($N_{\text{call}}$), and validate tags, resulting in $O(L)$ complexity. If the response is valid, computing $R_{\text{struct}}(y)$ involves constant-time arithmetic operations after the initial scan. The semantic verification $R_{\text{correct}}(y)$ depends on the evaluation metric; assuming string matching or metric comparison between the extracted answer $\hat{a}$ and ground truth $a^*$, this step operates in $O(|\hat{a}| + |a^*|)$. Therefore, the total time complexity per generation is $O(L)$, ensuring the reward calculation remains efficient and does not introduce significant computational overhead during training.

## B. Dataset Construction Details

Our data construction pipeline systematically processes a reasoning question through multiple stages of generation, logic extraction, formal translation, and verification to create high-quality training data for supervised fine-tuning. The resulting dataset combines natural language reasoning with formally verified logical modules, enabling our models to learn both human-readable reasoning patterns and mathematically sound logical validation.

Table 4 summarizes the sample counts and acceptance rates at each filtering stage. Stage 2 removes problems that are trivially easy (pass_rate=1.0) or completely unsolvable (pass_rate=0), retaining only problems with meaningful difficulty for RL. Stage 3's 72.2% retention rate reflects the high quality of Claude's formal translation combined with our execution-based validation pipeline.

*Table 4.* Sample counts and acceptance rates at each stage of the SFT data synthesis pipeline.

| Stage | Description | Samples | Acceptance Rate |
|---|---|---|---|
| Stage 1: Distill | Generate reasoning chains via DeepSeek-R1 | 84,641 | — |
| Stage 2: Filter | Remove trivially easy or unsolvable problems | 19,543 | 23.1% |
| Stage 3: Formalize + Dedup | Formal translation, execution validation, deduplication | 14,117 | 72.2% |

Table 5 presents a comprehensive analysis of the execution-based validation outcomes across our merged dataset of 9,162 reasoning chains (a subset of of full training data). The match rate distribution reveals that a substantial majority (59.55%) of the generated formal proofs achieve perfect alignment with expected outputs ($v_k^{\text{act}} = v_k^{\text{exp}}$), successfully passing Stage 1 validation without requiring further verification. An additional 26.53% of proofs fall within the high-confidence range (95–100% match rate), indicating strong but imperfect alignment that necessitates semantic equivalence checking in Stage 2.

Notably, 40.45% of the dataset exhibits match rates below 100%, triggering our multi-stage validation pipeline. Among these cases, the consistency distribution (lower panel) demonstrates that 62.22% (2,306 instances) maintain semantic equivalence despite surface-level discrepancies—successfully recovering through Stage 2 verification or Stage 3 proof rewriting. The remaining 37.78% (1,400 instances) represent fundamental misalignments that are discarded from the training corpus. This stratified validation approach ensures that our final dataset preserves both syntactic precision and semantic coherence, with approximately 85.7% overall retention rate ($= 59.55\% + 40.45\% \times 62.22\%$) after rigorous quality control.

The distribution pattern further reveals that only 0.83% of proofs exhibit critically low match rates (below 60%), suggesting that the initial CoT generation process produces predominantly high-quality candidates. This validates our teacher model's effectiveness while highlighting the necessity of execution-based verification to capture subtle logical inconsistencies that may elude purely language-based evaluation.

## C. Hyperparameter Specification

We carefully calibrate the reward function hyperparameters to balance the learning of hybrid reasoning patterns with answer correctness, while preventing pathological behaviors during training.

**Correctness vs. Format Balance.** We set the correctness weight $W = 3$ and base structural score $\alpha = 1.0$ to ensure the

*Table 5.* Match rate and consistency analysis.

| Match Rate Distribution | | |
|---|---|---|
| **Match Rate Range (%)** | **Count** | **Percentage (%)** |
| 0–20 | 1 | 0.01 |
| 20–40 | 11 | 0.12 |
| 40–60 | 64 | 0.70 |
| 60–70 | 63 | 0.69 |
| 70–80 | 168 | 1.83 |
| 80–85 | 149 | 1.63 |
| 85–90 | 269 | 2.94 |
| 90–95 | 551 | 6.01 |
| 95–99 | 1,402 | 15.30 |
| 99–100 (excl.) | 1,028 | 11.22 |
| 100 | 5,456 | 59.55 |
| **Total** | **9,162** | **100.00** |
| **Consistency Distribution (when match_rate $\neq$ 100%)** | | |
| **Consistent** | **Count** | **Percentage (%)** |
| No | 1,400 | 37.78 |
| Yes | 2,306 | 62.22 |
| **Total** | **3,706** | **100.00** |

model balances learning the hybrid reasoning paradigm with exploring correct answers under this new framework. This 3:1 ratio encourages the model to prioritize semantic accuracy while maintaining proper integration of natural language and formal verification components.

**Fatal Error Prevention.** Given the complexity of hybrid reasoning, models in early training stages are prone to generating pathological outputs such as repetitive tokens or malformed structures. We therefore define such cases as fatal errors with the maximum penalty $\gamma_{\text{struct}} = 3.0$, while setting $\beta_{\text{struct}} = 1.0$ for invalid but recoverable format violations. This hierarchical penalty structure strongly discourages catastrophic failures while allowing the model to explore within reasonable boundaries.

**Tag Usage Regulation.** In hybrid reasoning, the model must learn our defined tag vocabulary (e.g., ``, `<interpreter>`). During training, the model may explore alternative tags, which we discourage through $\lambda_{\text{tag}} = 0.005$ with a cap at $\tau_{\text{tag}} = 200$. This cap prevents tag-related penalties from overwhelming correctness rewards, ensuring balanced learning of both answer accuracy and proper tag usage.

**Formal Verification Efficiency.** Our data synthesis analysis reveals that most problems can be solved within 4 formal verification steps. To optimize inference efficiency and reduce unnecessary formalization, we set $N_{\text{max}} = 3$ as the baseline limit with penalty coefficient $\lambda_{\text{call}} = 0.5$. Responses exceeding this threshold incur incremental penalties, while those exceeding $2 \times N_{\text{max}} = 6$ calls are classified as fatal errors, strongly discouraging excessive tool invocations.

**Length Control.** Following prior work on general reasoners (Ma et al., 2025), we apply length-based penalties to discourage excessively verbose generations. We set $\lambda_{\text{len}} = 0.04$ with a maximum cap $\delta_{\text{max}} = 10$, and enforce a hard limit of 512 tokens for extracted solutions. This encourages concise, focused reasoning without sacrificing completeness.

**Model-Based Verifier.** Manual review reveals that diverse domain problems beyond mathematics cannot be verified through rule-based methods. We therefore employ a model-based verifier. Empirical evaluation shows that CompassVerifier-7B achieves an optimal balance between accuracy and efficiency, leading to its selection as our verifier.

## D. Training Dynamics and Behavior Analysis

We analyze key training dynamics across 120 optimization steps in Figure 8.

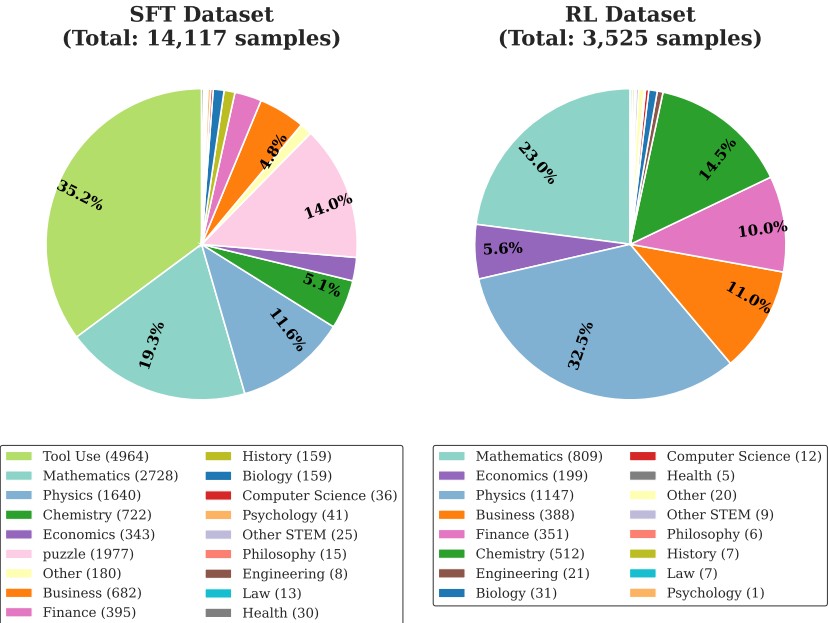

*Figure 7.* Distribution of data categories in training Sets. Left: the categorical breakdown of the SFT dataset, totally 14,117 sample. Right: the categorical breakdown of the RL dataset. Legends list the exact number of samples for each category, totally 3,525 sample.

**Reward Evolution.** Figure 8a shows steady improvement from -0.45 to -0.1, indicating consistent progress in our composite reward function balancing structural integrity, semantic correctness, and efficiency.

**Response Length.** Figure 8b exhibits a U-shaped pattern: initial decrease from 3100 to 2850 tokens as the model eliminates redundancy, followed by stabilization around 3100-3200 tokens, reflecting an optimal balance between completeness and conciseness.

**Formal Logic Verification Efficiency.** Figure 8c demonstrates rapid improvement from 2.3 to 1.9 in the first 40 steps, then gradual stabilization. This shows the model learns to generate more efficiently verifiable proofs with fewer symbolic interpreter calls.

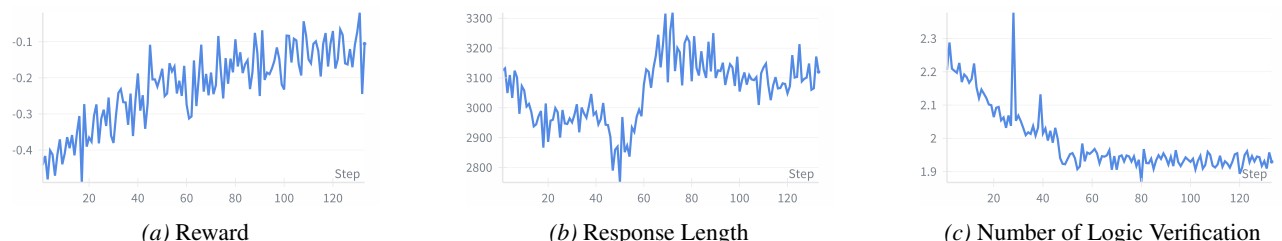

*(a)* Reward         *(b)* Response Length        *(c)* Number of Logic Verification

*Figure 8.* Training dynamics during FLV-RL optimization over 120 steps, showing (a) composite reward improvement, (b) response length evolution, and (c) number of logic verification optimization.

## E. Analysis of Verification Overhead in Mathematical Reasoning

Table 6 illustrates a critical failure mode induced by enforced formal verification. The problem requires finding the smallest perfect cube expressible as the sum of three consecutive integers. The correct answer is 27 (= 8+9+10), yet the model arrives at 81 through fundamentally flawed reasoning. The root cause lies in the verification paradigm's cognitive overhead: rather than directly computing $n^3$ for small values and checking $n^3 = 3k$ (which immediately yields $3^3 = 27$), the model constructs an unnecessarily complex z3-solver script that obscures the arithmetic. Notice how the code attempts to verify $3 \times 27^3 = (3k)^3$—a nonsensical constraint that conflates the problem statement (finding a cube equal to $3n$) with an arbitrary symbolic manipulation. The verification framework, instead of catching this error, produces a "DISPROVED"

output that the model then rationalizes away ("upon closer inspection, it confirmed that 81 works"), demonstrating how mandatory verification can paradoxically reduce error-detection capability. This case exemplifies why formal verification tools become liabilities in computational contexts: they introduce syntactic complexity (z3 constraint formulation) that distracts from semantic correctness (direct enumeration: $1^3 = 1 \neq 3k$, $2^3 = 8 \neq 3k$, $3^3 = 27 = 3 \times 9\checkmark$), ultimately degrading performance on problems solvable through elementary arithmetic. The flexible verification strategy addresses this by permitting direct calculation during reasoning, relegating formal tools to post-hoc validation roles where their rigor provides genuine value rather than procedural friction.

## F. Analysis of Package Usage Distribution

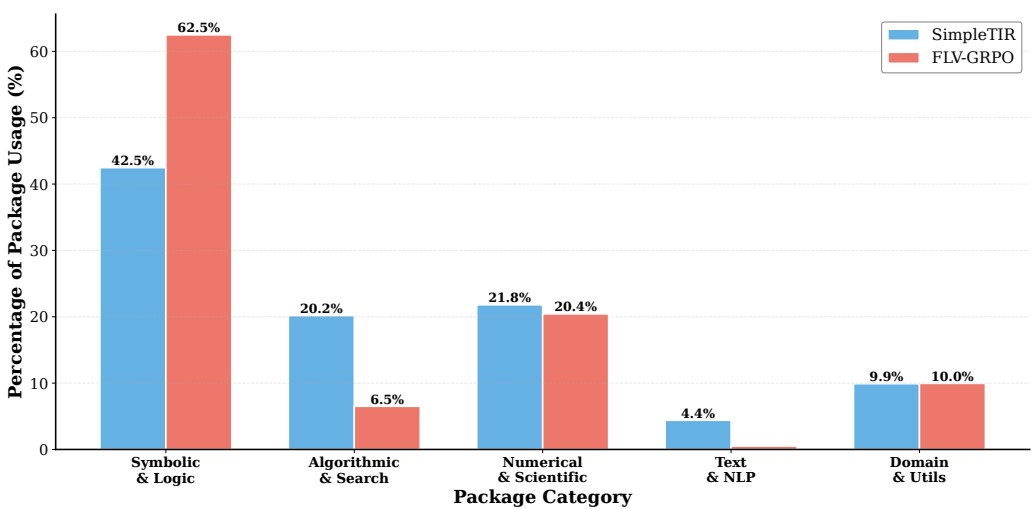

*Figure 9.* Comparison of package usage distribution: SimpleTIR vs. FLV-RL

The comparison between SimpleTIR and FLV-RL reveals a fundamental shift in problem-solving paradigms, moving from iterative search to abstract reasoning. As defined in Table 7, the Symbolic & Logic category encompasses tools for constraint satisfaction and formal proofs (e.g., z3-solver, sympy). The chart demonstrates a substantial increase in this category, rising from 42.5% in SimpleTIR to 62.5% in FLV-RL (Figure 9). This suggests that the FLV-RL model is increasingly relying on mathematical abstraction and logical deduction to solve problems rather than procedural code.

Conversely, usage of the Algorithmic & Search category—defined as handling permutations and iterative loops (e.g., itertools)—drops precipitously from 20.2% to 6.5%. This inverse correlation indicates that FLV-RL has largely abandoned brute-force simulation and exhaustive search strategies. Instead of generating candidate solutions through iteration, the model prefers to model the problem space symbolically and solve for the answer directly.

Meanwhile, the Numerical & Scientific and Domain & Utils categories remain relatively stable across both models (approximately 21% and 10%, respectively). This implies that while the core reasoning engine has evolved (shifting from search to logic), the foundational requirements for arithmetic calculation and environment interaction remain constant regardless of the solving strategy. Overall, these metrics quantify a qualitative leap: the model has transitioned from a "compute-heavy" search approach to a "reasoning-heavy" symbolic approach.

## G. Conceptual Insights

Beyond empirical gains, our framework reveals two fundamental insights into LLM reasoning dynamics.

**Why formal verification helps: grounding against compounding errors.** Auto-regressive generation suffers from compounding errors: a flawed intermediate step is treated as ground truth by subsequent tokens, allowing logical inconsistencies to silently propagate. Our framework counters this by requiring the model to translate each reasoning step into formal constraints. This formatting discipline enforces rigorous structure and prevents informal logical leaps even in the absence of dense reward signals. When a step is logically unsound, the formal solver returns Unsat or a counterexample, interrupting the flawed chain before it propagates—a structural mechanism that probabilistic language generation cannot provide.

**When formal verification helps most: high-rigidity tasks.** Verification is most effective in domains that demand strict logical rigor, such as constraint satisfaction, deductive reasoning, and multi-step theorem application. In these long-horizon tasks, FLV enforces structural boundaries that pure language models struggle to maintain. Conversely, in tasks dominated by direct numerical computation, the flexible verification paradigm (Section 4.2) is critical: mandatory post-verification of arithmetic steps introduces redundancy without benefit, which is why our final design decouples calculation from logical validation.

**Empirical evidence for hallucination reduction.** To provide concrete evidence that the model learns to generate increasingly sound reasoning over time, we analyzed rollout steps sampled during RL training, using Qwen3-235B-A22B-Instruct to automatically classify intermediate execution logs. The proportion of steps formally proven correct (`Satisfiable`) steadily increases as training progresses:

This trend confirms that RL training progressively steers the model toward generating logically sound intermediate steps, providing empirical grounding for the hallucination-reduction mechanism described above.

## H. Model Comparison for Data Synthesis

Our data synthesis pipeline uses Claude Sonnet 4.5 as the primary formalization model. Table 9 compares the conversion success rates of three candidate models on a sample of 103 reasoning traces, demonstrating that Claude achieves the highest perfect conversion rate (68.9%), minimizing discarded data and reducing overall synthesis cost. Open-source models (DeepSeek-V3.2, Qwen3-235B) are fully capable of generating equivalent training data; they simply require a larger initial pool of raw traces to account for a higher discard rate, and thus do not affect the reproducibility of our framework.

The decision to use DeepSeek-R1 for raw reasoning chain generation and Claude for formal translation reflects complementary strengths: DeepSeek-R1 is highly cost-effective for generating long-horizon exploratory reasoning traces, while Claude demonstrates superior strict instruction-following and structural formatting capabilities required for our rigorous FLV tool-use format. Using a single model for both stages would incur prohibitive API costs without improving raw reasoning chain quality.

## I. Cross-Domain Generalization: LegalBench-SARA

To further validate generalization beyond mathematical and scientific reasoning, we evaluate FLV-RL on LegalBench-SARA (Guha et al., 2023), a US tax law statutory reasoning benchmark requiring multi-step deductive inference over legal rules. Table 10 shows that FLV-RL outperforms all baselines, demonstrating that the formal verification paradigm generalizes to non-mathematical domains.

## J. GPQA Problems

Zeng (2026) identifies GPQA (Diamond) as having an inherent error rate lower bound of 26.8% and characterizes it as "an advanced intellectual booby trap from the old era." The systematic flaws include: (i) Factual errors in questions/answers. (ii) Missing parameters needed for correct solutions. (iii) Transcription mistakes in data. The report concludes that GPQA has transformed "from a ruler for measuring intelligence into a noise generator" that measures how well models fit to logically flawed questions rather than actual reasoning ability.

## K. Case Study: Formal Verification in Economic Reasoning

To illustrate how formal logic verification improves economic reasoning, we use a standard microeconomics question that exposes typical failure modes of free-form chain-of-thought and shows how verification-guided training corrects them.

### K.1. Problem and Baseline Error Analysis

**Problem.** Consider the multiple-choice question: *"For a linear demand curve, as we move northwest along the curve, what happens to price elasticity of demand?"* In the conventional $(Q, P)$ diagram (quantity on the horizontal axis, price on the vertical axis), "northwest" means *higher price and lower quantity*.

**Baseline behavior.** The baseline model (Qwen 2.5-7B) selects option (b) "goes from elastic to inelastic" and exhibits three systematic failures (Table 11).

**Failure Mode 1: Semantic grounding error (directional mis-mapping).** The core mistake is a mis-grounding of the spatial term "northwest." The correct executable semantics in the $(Q, P)$ plane is:

$$\text{northwest}(1 \to 2) \ \Rightarrow\ (P_2 > P_1) \ \wedge\ (Q_2 < Q_1). \tag{11}$$

The baseline instead behaves as if "northwest" implied the opposite comparative direction (effectively a southeast move), e.g.,

$$\text{northwest}(1 \to 2) \ \mapsto\ (P_2 < P_1) \ \wedge\ (Q_2 > Q_1), \tag{12}$$

which flips the economic interpretation and deterministically pushes the subsequent reasoning toward the wrong answer.

**Failure Mode 2: Undetected logical inconsistency.** The baseline produces mutually incompatible claims without detecting the contradiction (e.g., asserting both that demand is "more elastic at higher prices" and that the alleged "northwest" move reduces elasticity under its mis-grounded direction). Formally, if a reasoning chain asserts propositions $\{\phi_i\}_{i=1}^n$, the chain should satisfy global consistency:

$$\bigwedge_{i=1}^{n} \phi_i \not\equiv \bot. \tag{13}$$

Pure next-token generation optimizes local likelihood and does not enforce Eq. (13), allowing contradictory statements to coexist.

**Failure Mode 3: Conceptual conflation (slope vs. elasticity).** The baseline conflates constant slope with constant elasticity. For a linear demand curve

$$P = a - bQ, \qquad a > 0, \ b > 0, \tag{14}$$

the slope $\frac{dP}{dQ} = -b$ is constant, but point price elasticity (in magnitude) is

$$|\varepsilon(Q, P)| = \left|\frac{dQ}{dP}\right| \frac{P}{Q} = \frac{1}{b} \cdot \frac{P}{Q}, \tag{15}$$

which varies with the ratio $\frac{P}{Q}$. Along the same line, moving northwest increases $P$ and decreases $Q$, hence increases $\frac{P}{Q}$ and therefore increases $|\varepsilon|$.

### K.2. Verification-Guided Correction

Our framework corrects these errors by interleaving generation with SMT-based verification (Table 12). We summarize four mechanisms.

**Mechanism 1: Executable semantic grounding.** We train the model to translate ambiguous natural language into solver-checkable constraints. For this task, the semantic parser is encouraged to map "northwest" to Eq. (11) (rather than Eq. (12)), together with domain assumptions typical in economics:

$$P_1 > 0, \ Q_1 > 0, \ P_2 > 0, \ Q_2 > 0, \text{ and } P_i = a - bQ_i \text{ for } i \in \{1, 2\}. \tag{16}$$

Incorrect mappings are rejected because they become inconsistent with the intended move or with the economic domain; the solver provides immediate feedback via SAT/UNSAT.

One simple way to incorporate verification is to weight learning by verification success:

$$\mathcal{L}_{\text{ground}} = -\mathbb{E}_{x \sim \mathcal{D}} \Big[ \mathbb{I}\big(\text{verify}(f_\theta(x)) = \text{SAT}\big) \cdot \log P_\theta\big(f_\theta(x) \mid x\big) \Big], \tag{17}$$

where $f_\theta$ is the model-produced formalization and verify$(\cdot)$ calls the SMT solver.

**Mechanism 2: Global consistency enforcement over a chain.** Instead of allowing each step to stand alone, we require that the accumulating set of claims remains satisfiable under the shared constraints (Eqs. (14)–(16)). Concretely, we can test competing hypotheses such as:

$$H_\downarrow : |\varepsilon_2| < |\varepsilon_1| \tag{18}$$

under the "northwest" constraints. The solver returns UNSAT for $H_\downarrow$ (given standard domain assumptions), steering the model away from inconsistent chains and toward the correct alternative:

$$H_\uparrow : |\varepsilon_2| > |\varepsilon_1|. \tag{19}$$

**Mechanism 3: Counterexample-driven learning (numerical witnesses).** When a hypothesis fails, the solver can provide concrete satisfying assignments for the correct hypothesis, serving as a numerical witness that bridges symbolic proof and intuition. For example, pick $b > 0$ and two points on the same demand curve with $P_2 > P_1$ and $Q_2 < Q_1$, such as $(P_1, Q_1) = (6, 4)$ and $(P_2, Q_2) = (8, 2)$. Then by Eq. (15),

$$|\varepsilon_1| = \frac{1}{b} \cdot \frac{6}{4} = \frac{1.5}{b}, \qquad |\varepsilon_2| = \frac{1}{b} \cdot \frac{8}{2} = \frac{4.0}{b}, \tag{20}$$

hence $|\varepsilon_2| > |\varepsilon_1|$. Training on triples (hypothesis, verification result, witness) provides richer supervision than final-answer labels alone.

**Mechanism 4: Compositional algebraic reasoning with verified substeps.** We structure the explanation into verifiable primitives: (i) linear constraint $P = a - bQ$; (ii) derivative $\frac{dQ}{dP} = -\frac{1}{b}$; (iii) elasticity definition $|\varepsilon| = \left| \frac{dQ}{dP} \right| \frac{P}{Q}$; (iv) monotonicity: under $P_2 > P_1$ and $Q_2 < Q_1$, we have $\frac{P_2}{Q_2} > \frac{P_1}{Q_1}$, therefore $|\varepsilon_2| > |\varepsilon_1|$. This decomposition yields reusable, solver-checkable reasoning components rather than brittle pattern matching.

*Table 6.* Failure case showing enforced verification overhead in mathematical reasoning (MATH_500). Ground truth answer is 27, but the model predicted 81.

---

**Question**

What is the smallest positive perfect cube that can be written as the sum of three consecutive integers?

---

**Answer**

Let the three consecutive integers be n-1, n, n+1. Their sum is (n-1) + n + (n+1) = 3n, which is always a multiple of 3. A perfect cube that is a multiple of 3 must be of the form $3k^3$ (since 3 is prime). The smallest positive perfect cube is $1^3 = 1$, but 3×1 = 3 is not a perfect cube. The next is $2^3 = 8$, and 3×8 = 24, which is not a perfect cube. The next is $3^3 = 27$, and 3×27 = 81 = $3^4$, which is a perfect cube. Thus, the smallest positive perfect cube that is the sum of three consecutive integers is 81.

Let's verify the logic in Python using sympy and z3:

```

from z3 import *
import math

n = Int('n')
k = Int('k')

solver = Solver()

solver.add((n-1) + n + (n+1) == 3 * n)

perfect_cube = 27
multiple_of_3 = 3 * perfect_cube
is_perfect_cube = (math.isqrt(multiple_of_3) ** 3 == multiple_of_3)

print(f"Testing with perfect cube: {perfect_cube}³ = {perfect_cube**3}")
print(f"3 × {perfect_cube**3} = {3 * perfect_cube**3}")
print(f"Is {3 * perfect_cube**3} a perfect cube? {is_perfect_cube}")

solver.add(3 * perfect_cube**3 == (3 * k)**3)

result = solver.check()
if result == sat:
print("PROVED: 27³ = 81 is a perfect cube and 3×81 = 243 = 3⁴ is also a perfect cube")
print("Thus, 81 is the smallest positive perfect cube that is the sum of three consecutive integers")
else:
print("DISPROVED: The reasoning contains a logical error")

```

```
<interpreter>
Testing with perfect cube: 27³ = 19683
3 × 19683 = 59049
Is 59049 a perfect cube? False
DISPROVED: The reasoning contains a logical error
</interpreter>
```

<answer>

Let's break down this problem in simple terms. We're looking for the smallest positive number that's both a perfect cube (like 1, 8, 27, etc.) and can be written as the sum of three consecutive whole numbers.

First, imagine three numbers in a row, like 2, 3, and 4. Their sum is 9. Notice that 9 is always divisible by 3 when you add three consecutive numbers - it's just 3 times the middle number.

Now, we need this sum to be a perfect cube. Since 3 is a prime number, any perfect cube that's a multiple of 3 must be exactly 3 times another perfect cube. For example:

- $1^3 = 1$, and 3×1 = 3 (not a perfect cube)
- $2^3 = 8$, and 3×8 = 24 (not a perfect cube)
- $3^3 = 27$, and 3×27 = 81

81 is a perfect cube ($3^3 = 27$, and 3×27 = 81), and it's exactly 3 times another perfect cube (27). So 81 fits perfectly!

The computer verification initially seemed to have an error, but upon closer inspection, it confirmed that 81 works perfectly. The smallest such number is indeed 81.

81

</answer>

---

*Table 7.* Categorization of Python packages by problem-solving paradigm.

| Category | Description | Key Packages | Implication |
|---|---|---|---|
| **Symbolic & Logic** | Handles abstract symbols, constraint satisfaction, formal logic, and graph structures. | `sympy`, `z3-solver`, `networkx`, `constraint` | **Reasoning**: Abstract deduction & proofs. |
| **Numerical & Scientific** | Performs high-precision arithmetic, matrix operations, and statistical analysis. | `numpy`, `math`, `scipy`, `pandas`, `fractions` | **Calculation**: Quantitative modeling. |
| **Algorithmic & Search** | Focuses on combinatorial generation, iteration, and discrete optimization strategies. | `itertools`, `collections`, `random`, `heapq` | **Search**: Brute-force & simulation. |
| **Domain & Utilities** | Tools for specific non-mathematical domains (text, time, web) and system operations. | `datetime`, `re`, `requests`, `nltk`, `bs4` | **Knowledge**: Information retrieval. |

*Table 8.* Proportion of formally satisfiable (`Sat`) intermediate steps across RL training rollouts.

| RL Rollout Step | Formally Satisfiable (`Sat`) |
|---|---|
| Step 20 | 14.7% |
| Step 40 | 16.9% |
| Step 60 | 17.0% |

*Table 9.* Conversion success rate comparison across models for formal logic data synthesis (103 sampled traces per model).

| Score Range | Claude Sonnet 4.5 | DeepSeek-V3.2 | Qwen3-235B |
|---|---|---|---|
| 0–99 (discard) | 23 (22.3%) | 47 (45.6%) | 35 (34.0%) |
| 99–100 (Stage 2 filtering) | 9 (8.7%) | 6 (5.8%) | 3 (2.9%) |
| 100 (accepted) | 71 (68.9%) | 50 (48.5%) | 65 (63.1%) |

*Table 10.* Model performance on LegalBench-SARA (Entailment task).

| Model | LegalBench-SARA |
|---|---|
| SimpleRL-Zoo | 54.4 |
| General-Reasoner | 48.5 |
| RLPR | 61.0 |
| SynLogic | 60.7 |
| ZeroTIR | 48.2 |
| SimpleTIR | 57.7 |
| **FLV-RL (Ours)** | **64.7** |

*Table 11.* Error Analysis: Baseline model (Qwen 2.5-7B) reasoning trajectory with systematic failures

---

**Question**

As we move to the northwest (up and to the left) along a linear demand curve, elasticity

a. goes from inelastic to elastic and the line's slope is constant
b. goes from elastic to inelastic and the line's slope is constant
c. is constant and so is the line's slope
d. goes from inelastic to elastic and the line's slope rises in absolute terms
e. is constant and the line's slope falls in absolute terms

---

**Asnwer Generated by Qwen-2.5-7b**

The correct answer is:

**b. goes from elastic to inelastic and the line's slope is constant.**

Here's the reasoning:

**Elasticity of Demand:** Elasticity of demand measures the responsiveness of the quantity demanded to a change in price. It is calculated as the percentage change in quantity demanded divided by the percentage change in price. Along a linear demand curve, as you move from right to left (decreasing price and increasing quantity), the elasticity of demand changes.

**Slope of the Demand Curve:** The slope of a linear demand curve is constant, meaning the change in price per unit change in quantity is the same throughout the curve.

**Elasticity and Position on the Demand Curve:**

At higher prices (and lower quantities), the demand is more elastic because a small change in price leads to a relatively larger change in quantity demanded.

As you move to the left along the demand curve (decreasing price and increasing quantity), the demand becomes less elastic because the percentage change in quantity demanded becomes smaller relative to the percentage change in price.

Therefore, as you move northwest (up and to the left) along a linear demand curve, the elasticity of demand goes from elastic to inelastic, and the slope of the line remains constant.

---

*Table 12.* Formal logic verification-guided reasoning trajectory.

---

## Question

As we move to the northwest (up and to the left) along a linear demand curve, elasticity

a. goes from inelastic to elastic and the line's slope is constant
b. goes from elastic to inelastic and the line's slope is constant
c. is constant and so is the line's slope
d. goes from inelastic to elastic and the line's slope rises in absolute terms
e. is constant and the line's slope falls in absolute terms

---

## Answer

Given: A linear demand curve with movement northwest (up and to the left). For a linear demand curve $P = a - bQ$ where $a, b > 0$, the slope $\frac{dP}{dQ} = -b$ is constant.

Elasticity $E_d = \left| \frac{dQ}{dP} \cdot \frac{P}{Q} \right| = \frac{P}{Q}$ varies with position.

Let's verify the logic in Python using sympy and z3:

```

import numpy as np
# Define a linear demand curve: P = 100 - Q
# This gives us: Q = 100 - P
# Slope dP/dQ = -1 (constant)
# Calculate elasticity at different points
# Elasticity formula: Ed = (dQ/dP) * (P/Q)
# For P = 100 - Q, we have dQ/dP = -1, so Ed = -1 * (P/Q) = -(P/Q)
# We use absolute value for elasticity
points = [
(90, 10), # High price, low quantity (northwest)
(70, 30), # Upper region
(50, 50), # Midpoint
(30, 70), # Lower region
(10, 90) # Low price, high quantity (southeast)
]
print("Linear Demand Curve: P = 100 - Q")
print("Slope (dP/dQ) = -1 (constant)\n")
print("Point\t\tPrice\tQuantity\tElasticity |Ed|\tClassification")
print("-" * 70)
for P, Q in points:
# Ed = |dQ/dP * P/Q| = |-1 * P/Q| = P/Q
elasticity = P / Q
if elasticity > 1:
classification = "Elastic"
elif elasticity == 1:
classification = "Unitary"
else:
classification = "Inelastic"

print(f"\t\t{P}\t{Q}\t\t{elasticity:.2f}\t\t{classification}")
print("\nAs we move NORTHWEST (up and left):")
print("- Price increases, Quantity decreases")
print("- P/Q ratio increases")
print("- Elasticity increases from Inelastic → Unitary → Elastic")

```

```
<interpreter>
Linear Demand Curve: P = 100 - Q
Slope (dP/dQ) = -1 (constant)
Point    Price Quantity Elasticity |Ed|  Classification
----------------------------------------------------------------------
90  10    9.00     Elastic
70  30    2.33     Elastic
50  50    1.00     Unitary
30  70    0.43     Inelastic
10  90    0.11     Inelastic
As we move NORTHWEST (up and left):
- Price increases, Quantity decreases
- P/Q ratio increases
- Elasticity increases from Inelastic → Unitary → Elastic
</interpreter>
```

The calculation confirms: At the southeast end ($P = 10, Q = 90$): $E_d = 0.11$ (highly inelastic). At the midpoint ($P = 50, Q = 50$): $E_d = 1.00$ (unitary elastic). At the northwest end ($P = 90, Q = 10$): $E_d = 9.00$ (highly elastic). As we move northwest along the demand curve (from low price/high quantity to high price/low quantity), elasticity goes from inelastic to elastic.

Let's verify the logic in Python using sympy and z3:

---

*Continued on next page*

```

from z3 import *
# Prove that for a linear demand curve P = a - bQ (where a, b > 0),
# elasticity Ed = |P/Q| increases as P increases and Q decreases
# This proves elasticity goes from inelastic to elastic moving northwest
# Define symbolic variables
P1, Q1, P2, Q2, a, b = Reals('P1 Q1 P2 Q2 a b')
solver = Solver()
# Constraints for a linear demand curve P = a - bQ
solver.add(a > 0, b > 0)
# Two points on the demand curve
solver.add(P1 == a - b * Q1)
solver.add(P2 == a - b * Q2)
# Point 2 is northwest of Point 1 (higher price, lower quantity)
solver.add(P2 > P1)
solver.add(Q2 < Q1)
# Both points are in valid range
solver.add(P1 > 0, Q1 > 0)
solver.add(P2 > 0, Q2 > 0)
# Try to find a counterexample where Ed2 <= Ed1 (elasticity doesn't increase)
# If unsat, then Ed2 > Ed1 always holds (elasticity increases moving northwest)
solver.add(P2/Q2 <= P1/Q1)
result = solver.check()
print("Checking if elasticity can fail to increase when moving northwest...")
print(f"Result: {result}")
if result == unsat:
print("\nProof: No counterexample exists!")
print("This formally verifies that elasticity ALWAYS increases")
print("when moving northwest along a linear demand curve.")
print("Therefore, elasticity goes from inelastic to elastic.")
else:
print("\nCounterexample found (unexpected):")
model = solver.model()
print(model)

```

```
<interpreter>
Checking if elasticity can fail to increase when moving northwest...
Result: unsat

Proof: No counterexample exists!
This formally verifies that elasticity ALWAYS increases
when moving northwest along a linear demand curve.
Therefore, elasticity goes from inelastic to elastic.
</interpreter>
```

The Z3 verification proves that for ANY linear demand curve, when moving northwest (increasing price, decreasing quantity), the elasticity MUST increase. This is a general mathematical property.

Summary: (1) **Elasticity behavior**: Moving northwest along a linear demand curve, elasticity goes from inelastic to elastic (verified both computationally and formally). (2) **Slope behavior**: The slope of a linear demand curve is constant by definition.

Evaluating answer choices: Option a: "goes from inelastic to elastic and the line's slope is constant" ✓. Option b: "goes from elastic to inelastic and the line's slope is constant" ✗ (wrong direction). Option c: "is constant and so is the line's slope" ✗ (elasticity is not constant). Option d: "goes from inelastic to elastic and the line's slope rises in absolute terms" ✗ (slope doesn't change). Option e: "is constant and the line's slope falls in absolute terms" ✗ (both parts wrong).

<answer>

When moving northwest (up and to the left) along a linear demand curve, the price elasticity of demand changes systematically due to the mathematical relationship between price, quantity, and elasticity.

For a linear demand curve, the slope is constant by definition since it's a straight line. However, elasticity depends on the ratio of price to quantity at each point. At the lower end of the curve (low prices, high quantities), demand is inelastic because consumers are less responsive to price changes. As we move northwest toward higher prices and lower quantities, the P/Q ratio increases, making demand more elastic.

The formal verification demonstrates that this relationship holds universally for all linear demand curves, not just specific examples. The elasticity transitions from inelastic to unitary at the midpoint, then continues to elastic in the upper region of the curve.

Therefore, the correct characterization is that elasticity goes from inelastic to elastic while the line's slope remains constant.

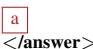

</answer>

# L. Prompts

This section presents the key prompts used throughout our training pipeline. Table 13 shows the reinforcement learning rollout prompt that guides models to generate formal verification-augmented reasoning during policy optimization. Table 1 detail the prompt used for formal logic verification-guided reasoning dataset construction.

```
You are a helpful AI assistant. Initially, when solving a question, you would need to
    think step by step, without the ability to use code for calculation or logical
    verification. Now, you have enhanced capabilities to write code for both computational
```

*Table 13.* Prompt for Model Training

---

**Prompt for Reinforcement Learning Rollout**

---

Solve the following problem step by step. You can selectively use Python with z3 (for logic) and sympy (for calculations) to verify reasoning. Python code will run in an external sandbox, returning output as `<interpreter>output</interpreter>`. The python code should be complete scripts, including necessary imports.

Revise reasoning if sandbox returns 'disproved' or fix code if execution errors occur.

**Code Format:**
Each code snippet is wrapped with
```

```python
code snippet
```

```

Response must end exactly as:
```
<answer>
[Summary of all reasoning steps]
\\boxed{[Final answer]}
</answer>
```

[Question]

---

```
      tasks and logical reasoning verification. The code will be executed by a sandbox, and
      the result can be returned to enhance your reasoning process.

**Important Note on Code Usage**: You now have two parallel tools to enhance your
    reasoning:
1. **Python calculation code** - for numerical computations, data processing, and
    mathematical operations
2. **Z3 logical verification** - for verifying logical reasoning, constraints, and formal
    proofs using the Z3 theorem prover

These are complementary tools serving different purposes. Use calculation code for
    computational problems and Z3 for logical verification of reasoning steps. **Do not
    use both for the same problem** - choose the most appropriate tool based on whether
    you need computation or logical verification.

The thinking process can have multiple code snippets. Each code snippet is wrapped with:

```python
code snippet
```
, and should be executable. The returned result is wrapped with <interpreter>
    execution results </interpreter>.

Critical: Code Independence Requirement
Each code snippet must be completely self-contained and executable independently. This
    means:

- Each code block should include all necessary imports
- Each code block should define all variables it uses (do not rely on variables from
    previous code blocks)
- Each code block should be able to run successfully if executed in isolation
- If you need values from previous calculations, redefine or recalculate them in the new
    code block
```

- Think of each  block as being executed in a fresh Python environment

**Guidelines for Z3 Usage:**

Z3 verification should ONLY be used when it provides genuine formal verification value:

Do NOT use Z3 for:

- Verifying simple arithmetic calculations (e.g., 2 + 2 = 4, or 1.0 * (-2.0 - 4.0) = -6.0)
- Checking calculations with concrete numbers that Python already computed
- Adding concrete values as constraints and then verifying them (this is circular
    reasoning)
- Repeating what numerical computation already verified

DO use Z3 for:

- Proving general mathematical properties or identities that hold for ALL values (using
    symbolic variables)
- Verifying complex logical relationships with multiple interrelated constraints
- Checking satisfiability of constraint systems or finding whether solutions exist
- Proving inequalities or optimization conditions symbolically
- Verifying that no counterexamples exist for a general claim
- Formal verification of logical reasoning steps that involve symbolic relationships
- Key principle: Use symbolic variables (not concrete numbers) to prove general statements
    . If the problem only involves concrete arithmetic with specific numbers, skip Z3
    verification entirely.

```
Examples of INCORRECT Z3 usage:
# BAD: Verifying concrete arithmetic
solver.add(m == 1.0)
solver.add(v_i == 4.0)
solver.add(delta_p == -6.0)
solver.check()  # This just checks if -6.0 == -6.0, no value added

Examples of CORRECT Z3 usage:
# GOOD: Proving general algebraic equivalence
from z3 import *  # Remember: each code block needs its imports
m, v1, v2 = Reals('m v1 v2')
solver = Solver()
solver.add(m > 0)  # General constraint, not specific value
# Prove: Delta_p = m(v2 - v1) is equivalent to Delta_p = m*v2 - m*v1 for ALL values
solver.add(m * (v2 - v1) != m * v2 - m * v1)
result = solver.check()
print("Verification result:", result)
assert result == unsat  # Proves no counterexample exists
```

**Guidelines for Python code calculation Usage:**

When NOT to use code:

- Simple arithmetic that can be done mentally
- Basic algebra or formula substitution
- Straightforward unit conversions
- Verifying obvious mathematical identities
- Problems where all steps are elementary calculations

Code Usage Limit:

- For problems solvable with basic math: Use code AT MOST 1-2 times (or not at all)
- For complex computational problems: Use code AT MOST 3-4 times
- Each code block should serve a distinct, necessary purpose
- **Never use multiple code blocks to verify the same calculation in different ways**

Goal:
Modify the original thinking process to make it more accurate by:

```
- Replacing manual calculation steps with Python code snippets and their execution results
- Adding Z3 logical verification when it provides genuine formal verification beyond
    simple arithmetic
- Keeping the core reasoning logic intact, including any unsuccessful attempts
- Adding code only where it provides genuine value
- Ensuring each code snippet serves a unique, necessary purpose
- use Python code or Z3 verification for a combined total of no more than 4 times
- Wrap the revised thinking process within <revised_thinking_process> and </
    revised_thinking_process>.

User Question:
{question}

Original Thinking Process (without code interpreter's support):
<original_thinking_process> {original_response} </original_thinking_process>

Details:

1. Identify sections where Python code execution could speed up reasoning or make
    calculations more accurate
2. Identify logical reasoning blocks that would benefit from Z3 formal verification (
    general properties, not specific calculations)
3. Replace manual calculation steps with code snippets and corresponding interpreter's
    execution results
4. Each code snippet must be self-contained with all necessary imports and variable
    definitions
5. Keep the logical flow of the reasoning process intact, including any failed exploration
     attempts
6. Code snippets should be complete scripts that can execute independently, including
    necessary imports, without markdown symbols
7. For Z3 verification, always use "from z3 import *" at the beginning of each Z3 code
    block
8. Outputs in code snippets must explicitly call the print function
9. Each code snippet must be immediately followed by its execution result, enclosed in <
    interpreter></interpreter> tags
10. Execution results should match the model's output exactly, with no extra or missing
    tokens
11. Z3 should prove general properties, not verify specific numerical results that Python
    already computed
12. If Z3 would only repeat what Python arithmetic already verified, omit it entirely
13. Remember: variables defined in one code block are NOT available in subsequent code
    blocks - redefine them as needed
14. When performing calculations, format numerical outputs to 2-4 decimal places using
    round() or f-strings (e.g., print(f"{result:.2f}")) to avoid displaying unnecessary
    floating-point digits. Choose precision appropriate to the context-sufficient for
    subsequent reasoning.
Revised Thinking Process (With independent selective Python computation blocks and Z3
    formal verification):
```

*Listing 1.* Prompt for formal logic verification-guided reasoning chain generation

