# OpenReview forum: "Pushing the Boundaries of Natural Reasoning: Interleaved Bonus from Formal-Logic Verification"
_ICML.cc/2026/Conference — ICML 2026 regular_

### Official Review · Reviewer_spzh · 2026-03-12

**Soundness:** 3
**Presentation:** 2
**Significance:** 3
**Originality:** 3
**Overall Recommendation:** 4
**Confidence:** 4

**Summary:**

The paper proposes FLV, a training framework that teaches LLMs to interleave natural-language reasoning with checkable steps (code, proof, verifier execution output) so mistakes can be caught and corrected during generation. It builds a large supervision set by distilling teacher solutions, split the reasoning into steps, translating each step into verifiable code, and filtering examples by verification/execution outcomes. Models are then trained with verification-guided SFT followed by GRPO using hierarchical rewards that enforce correct tool formatting/usage and final answer correctness. Evaluations on multiple reasoning benchmarks with Qwen2.5-7B/14B backbones show consistent accuracy gains over natural-language -SFT/-RL baselines.

**Compliance With Llm Reviewing Policy:**

Affirmed.

**Final Justification:**

I did not change my overall recommendation following the rebuttal.

I reduced the Significance score from 4 to 3 due to the limited depth of insights regarding formal guarantees.

**Key Questions For Authors:**

- What are the evaluation results of the teacher models --- especially the synthesizer (Claude-Sonnet-4.5)? Reporting teacher performance (and how it compares to the trained student) would help quantify how much headroom exists and how effectively distillation/transformation transfers capability.
- Missing/desired ablations:
  - RL without SFT: How does GRPO perform without the FLV-SFT "cold start"? Is SFT necessary, or mainly helpful for stability/sample efficiency?
  - Sensitivity to $\lambda_{len}$: Since Figure 5 suggests the method increases output length, can you report accuracy as a function of the length penalty/weight to show the trade-off?
  - Verifier-iteration budget: How does accuracy change with the number of iterative verification rounds (or tool calls)? This matters for practical deployment cost.
- What is the cost of synthesizing the FLV-SFT data? Please report approximate time cost, the acceptance rate, and how many candidate traces are rejected at each filtering stage.
- Why are multiple teacher models used (DeepSeek-R1 for distillation, GPT-4o as a judge, and Claude-Sonnet-4.5 for synthesis)? Should similar results be obtained with a single teacher?
- Why is the length reward integrated inside $R_{logic}$ rather than treated as a separate reward term?
- Presentation (Minor)
  - Figure 1 shows a case where the final answer is incorrect but intermediate steps are correct, which seems inverted relative to the stated motivation (correct final answer despite flawed intermediate reasoning). Can you clarify why this example is representative?
  - Figure 3 is not referenced clearly in the text, and its stages do not map cleanly to the described synthesis pipeline (e.g., steps 1–3 appear in the figure but not in the narrative). Please revise the exposition so readers can directly align the figure with the written procedure.

**Limitations:**

Despite the use of formal tools, the method does not provide end-to-end theoretical guarantees about correctness, which limits its suitability for safety-critical deployments. It is heuristic rather than sound: while verifier outputs can filter some errors, the overall generation-decomposition-formalization loop does not guarantee globally correct or complete supervision traces.

**Strengths And Weaknesses:**

**Strengths:**
- The topic is timely and highly relevant to AI alignment and AI safety, especially for reducing logically inconsistent reasoning.
- The evaluation is broad and covers multiple benchmarks, providing a reasonably comprehensive empirical picture.
- The approach is presented as domain-general, with experiments spanning logical, mathematical, and general reasoning tasks.

**Weaknesses:**
- The central idea largely amounts to incorporating verification tools into the data-generation and training loop; the paper does not offer deeper conceptual insights into when and why verification guidance helps, beyond empirical gains.
- Comparisons to stronger state-of-the-art models (including the teacher models used for distillation) are missing, and the ablation coverage could be stronger/clearer to isolate the contribution of each component (please see "Key Questions For Authors" below).
- The synthesis pipeline may be expensive: many generated traces are rejected, and these discarded queries represent wasted computation.
- The pipeline assumes access to a substantially stronger teacher model, which may not always be available.
- Presentation issues:
  - The benchmarks and datasets used for evaluation (e.g., WebInstruct-Verified, K&K, NuminaMath-TIR) are not explained clearly enough in the main text.
  - The “formal verification” component is under-specified: What are the verifier inputs (isn't a verification query specific in each domain)? What are its outputs ("False/True/Unknown"? How is "Unknown" handled)? What a verification trace looks like?
  - A running example carried through the pipeline would substantially improve clarity and help readers understand the method end-to-end.

---

> ### Author Rebuttal · Authors · 2026-03-31
>
> Dear Reviewer spzh,
>
> Thank you for your positive assessment and the constructive questions! Below, we provide a comprehensive response to the concerns and questions you raised.
>
> ---
> > **W1: Deeper conceptual insights.**
>
> Beyond empirical gains, our framework reveals fundamental insights into LLM reasoning dynamics:
>
> **Why it helps (Grounding):** Auto-regressive generation suffers from compounding errors. Formatting steps for verification forces rigorous structures, preventing informal logical leaps even without dense rewards.
>
> **When it helps (High-Rigidity Tasks):** Verification is most effective in domains that require rigious logic. In these long-horizon tasks, FLV enforces structural boundaries that pure language models struggle to maintain.
>
> We will add a dedicated "Conceptual Insights" section in the revised version to explicitly detail these dynamics.
>
> > **W2 & Q1: Teacher model performance.**
>
> We respectfully emphasize this is not a knowledge distillation paper. Claude acts strictly as a **data synthesis tool** to translate reasoning traces into our FLV format, not to teach reasoning.
>
> However, we agree comparing against the teacher demonstrates the tool-use capabilities in data synthesis. We evaluated Claude-Sonnet-4.5 under our protocol (max 4 tool calls) and will include these scores in furthre versions.
>
> **Table 1: Comparison between Teacher model and FLV (Ours) under the unified protocol.**
> | Model | KOR-Bench | AIME24 | GPQA-D |
> |--|--|--|--|
> | Claude Sonnet 4.5 | **67.8** | **72.7** | **77.3**
> | FLV-SFT (Ours) | 48.0 | 20.0 | 32.3
> | FLV-RL (Ours) | 51.0 | 20.8 | 35.4
>
> > **W3 / Q3: Synthesis pipeline cost, acceptance rate.**
>
> The pipeline takes ~2 days, mostly distilling reasoning chains from DeepSeek-R1. We constrained tool calls and optimized prompts to minimize waste. Acceptance rates and rejection statistics are detailed in Table 4, showing exact match, semantic check, and rewrite success rates yielding our final 17k samples.
>
> > **Q2: Missing/desired ablations:**
> - *RL without SFT*
>
> We agree this is an important baseline. Our experiments show that applying RL without SFT initialization (FLV-ZeroRL) fails to teach the model our "Formal Tool-as-Verifier" paradigm, yielding inadequate improvement (**base: 30.3, FLV-ZeroRL: 42.7, FLV-RL: 51.9**). We will include detailed results in the revised version.
>
> - *Sensitivity to λlen*
>
>     We adopted the λlen penalty directly from General-Reasoner's optimal configuration. Due to computational constraints, we cannot provide a full sensitivity analysis for λlen. We will clarify this hyperparameter adoption in the implementation details.
>
>     *Reference*
>
>     [1] Ma X, Liu Q, Jiang D, et al. General-reasoner: Advancing llm reasoning across all domains[J]. arXiv preprint arXiv:2505.14652, 2025.
>
> - *Verifier-iteration budget*
>
>     Verification rounds are capped at 4 (§5.1), as most corrections occur within this limit. We will add a distribution figure in revision.
>
> > **Q4: Multiple teacher models rationale.**
>
> Our work is not traditional knowledge distillation. We use LLMs purely as **data synthesis tools** to meet formal verification requirements, chosen for efficiency, formatting, and cost. We primarily utilized Claude 4.5 Sonnet due to its superior perfect conversion rate (68.9%), which minimizes discarded data and costs (prompts in Appx I).
>
> **Table 2. Conversion success rate comparision between different models.**
> | Range              | Claude 4.5 Sonnet | DeepSeek-V3.2   | Qwen3-235B-A22B-Instruct      |
> |--|--|--|--|
> | 0–99 (discard)             | 23 (22.3%)       | 47 (45.6%)      | 35 (34.0%)
> | 99–100 (Stage 2 filtering)     | 9 (8.7%)         | 6 (5.8%)        | 3 (2.9%)
> | 100 (accepted)                | 71 (68.9%)       | 50 (48.5%)      | 65 (63.1%)
>
> > **Q5: Why is λlen inside Rlogic rather than a separate term?**
>
> Coupling length efficiency with correctness (Eq. 9) penalizes only correct but verbose answers. A separate term could over-penalize necessary verification steps. Embedding λlen ensures length optimization remains secondary.
>
> > **Limitation: guarantees about correctness.**
>
> We acknowledge lacking absolute, end-to-end theoretical guarantees for safety-critical deployments.
>
> However, our contribution significantly advances empirical rigor. By interleaving flexible language with strict formal verification, FLV drastically reduces logical hallucinations versus standard LLMs. While partially heuristic, our approach bridges open-domain reasoning and formal logic—a crucial practical step toward fully verifiable AI.
>
> We will address all presentation issues in revision.
>
> ---
> Finally, we express our heartfelt gratitude for your insightful questions and suggestions. They have been invaluable to our work and will be integral to our revised manuscript. We hope our efforts will adequately address your concerns and that you will recognize our work.

---

> > ### Author Rebuttal · Reviewer_spzh · 2026-04-03
> >
> > The additional clarifications regarding the role of the teacher models, the synthesis pipeline cost, and the RL-without-SFT ablation help address several of my concerns.
> >
> > That said, I would appreciate further clarification on a few remaining points:
> >
> > *Teacher–student performance gap.*
> > The reported results show a substantial gap between the teacher (e.g., Claude Sonnet 4.5) and the trained student models. Do you have an explanation for this gap, and are there concrete directions to close it? In particular, if there is significant headroom, why does RL provide only incremental improvements over SFT?
> >
> > *Synthesis cost and efficiency.*
> > While the “~2 days” estimate is helpful, it would be valuable to better understand the efficiency of the pipeline. Could you provide clearer statistics on acceptance rates and how many candidate traces are rejected at each filtering stage?
> >
> > *Rationale for multiple teacher models.*
> > The response explains that different models are used for different roles (distillation, synthesis, judging), with Claude performing best as a synthesizer. However, it remains unclear why a single strong model (e.g., Claude) could not be used for all stages. Is there an empirical or cost-based reason for this design choice?

---

> > > ### Author Response · Authors · 2026-04-03
> > >
> > > Dear Reviewer spzh,
> > >
> > > We are glad that our previous response helped address several of your concerns, and we thank you for your constructive follow-up questions.
> > >
> > > > 1. Teacher–student performance gap.
> > >
> > > We appreciate your insightful observation regarding the performance gap. As emphasized earlier, we do not use Claude to distill its vast world knowledge or raw reasoning power. The formal translation process is primarily designed to capture Claude's interpretable reasoning and verify its correctness on the given data. While this added explanatory information enhances the data, it does not fully account for the inherent capability gap between Claude and our 7B student model. Ultimately, we strictly use Claude as an auxiliary formatting tool to help the smaller model learn this formal verification pattern. In this context, our work represents a first attempt to verify the feasibility of the "Formal Tool-as-Verifier" paradigm at a smaller, accessible scale, rather than aiming to match the teacher's absolute performance.
> > >
> > > This capacity constraint also explains RL's incremental gains over SFT. FLV-SFT successfully imparts the formal-verification format, bringing the 7B model near its capacity ceiling. While RL further optimizes logical trajectories and reduces hallucinations (outperforming other 7B baselines), it cannot synthesize missing domain knowledge.
> > >
> > > > 2. Synthesis cost and efficiency.
> > >
> > > We apologize if our previous reference to Table 4 was not explicit enough. To clarify the pipeline's efficiency, here are the concrete statistics for the traces processed at each stage for SFT data construction:
> > >
> > > | Stage | Original Samples | Remained Samples | Acceptance Rate |
> > > |---|---|---|---|
> > > | Stage 1: Distill | 84,641 | 84,641 | — |
> > > | Stage 2: Filter by pass rate | 84,641 | 19,543 | 23.1% |
> > > | Stage 3: Formalization + Dedup | 19,543 | 14,117 | 72.2% |
> > >
> > > Filter by pass rate is to remove problems that are trivially easy (pass_rate = 1.0) or unsolvable (pass_rate =0).
> > >
> > > We hope this detailed breakdown addresses your concern, and we will include this table in the revised version of the paper.
> > >
> > > > 3. Rationale for multiple teacher models.
> > >
> > > Thank you raising this valuable question. The decision to use different models is driven by both **empirical strengths and cost-efficiency**. DeepSeek-R1 is highly specialized (and cost-effective) for generating raw, exploratory, long-horizon reasoning chains. Claude, on the other hand, empirically demonstrates superior strict instruction-following and structural formatting capabilities, making it the ideal choice for translating raw text into our rigorous FLV tool-use format.
> > >
> > > Indeed, as your question suggests, using a single strong model like Claude for the entire pipeline—including the generation of massive, exploratory reasoning traces from scratch—would incur prohibitive API costs without necessarily improving the quality of the raw reasoning chains. By decoupling the reasoning generation (R1) from the format translation (Claude), we achieve a highly scalable and cost-effective synthesis pipeline.
> > >
> > > ---
> > >
> > > Thank you for your insightful questions, which inspired deeper reflection on our method's transferability and efficiency.

---

### Official Review · Reviewer_DG1B · 2026-03-12

**Soundness:** 2
**Presentation:** 2
**Significance:** 2
**Originality:** 1
**Overall Recommendation:** 3
**Confidence:** 3

**Summary:**

This paper proposes a formal logic verification (FLV) framework to improve the reasoning capabilities of large language models. The key idea is to integrate executable verification tools (e.g., Python for computation, the Z3 theorem prover for logical checking) into the model’s reasoning process. The authors first train the model via supervised fine-tuning to generate structured reasoning sequences that include reasoning steps, formal verification code, and execution results. They then apply reinforcement learning with a hierarchical reward function that evaluates format correctness, structural validity, and logical correctness.

**Compliance With Llm Reviewing Policy:**

Affirmed.

**Final Justification:**

The distinction between this work and existing literature on LLMs combined with formal tool-use is not sufficiently clear, as also noted by the other reviewers. The authors provided some valid clarifications, yet some of the distinctions remain merely implementation-based. Given the current state of the manuscript, I still lean toward rejection.

**Key Questions For Authors:**

1. What is the fundamental difference between the proposed approach and prior work that incorporates verifier tools into LLM reasoning?

2. The introduction of formal verification tools (e.g., Z3) is an interesting component of the framework. However, the paper does not provide a detailed analysis of the training time or computational overhead introduced by this design, nor does it compare the training efficiency with alternative approaches in the experiment section. This raises the question of how much additional cost the verification process introduces and whether the performance improvements justify the potential increase in computational expense.

**Limitations:**

Yes.

**Strengths And Weaknesses:**

The methodology is clearly described and appears technically sound. The experimental results are also reasonably strong.

However, combining LLMs with external tools such as Python or Z3 for reasoning has already been explored in prior work, e.g., [1-3]. The novelty of this work is therefore not very clearly demonstrated. Although the authors claim that their approach differs from previous methods by using formal verification as a verifier rather than as a solver and by integrating it into the training process, these differences appear to be largely at the level of system design or integration, and it remains somewhat unclear to what extent they constitute a fundamentally new methodological contribution.

The experiments mainly focus on mathematical and logical reasoning tasks, and it remains unclear how well the method generalizes to other types of reasoning problems. In addition, comparisons with some recent verifier-based reasoning or tool-integrated reasoning approaches are still limited.

[1] Ryu et al., Divide and Translate: Compositional First-Order Logic Translation and Verification for Complex Logical Reasoning. ICLR 2025.

[2] Sadeddine et al., Verifying the Steps of Deductive Reasoning Chains. ACL 2025 Findings.

[3] Feng et al., VeriCoT: Neuro-symbolic Chain-of-Thought Validation via Logical Consistency Checks. arXiv 2025.

---

> ### Author Rebuttal · Authors · 2026-03-31
>
> Dear Reviewer DG1B,
>
> Thank you for your assessment and the pointed questions! Below, we provide a comprehensive response to the concerns and questions you raised.
>
> ---
> > **W 1 & Q1: Fundamental difference from prior work on verifier-integrated LLM reasoning.**
>
> While combining LLMs with external tools is indeed well-explored, **FLV is not merely a system-level integration of tools; it introduces a fundamental methodological shift in how LLMs learn to reason.**
>
> **1. Bypassing Post-Hoc Autoformalization:** Methods like CLOVER [1] and VeriCoT [3] translate generated natural language into logic post-hoc, causing severe semantic alignment errors. FLV uses interleaved generation—synchronizing reasoning and verification step-by-step—entirely bypassing translation bottlenecks. (VANESSA [2] avoids this only via hard-coded formulas).
>
> **2. Intrinsic RL Grounding:** Standard reasoning RL suffers from unconstrained exploration and compounding hallucinations. Instead of filtering final answers, FLV embeds the verifier in-loop during RL rollouts. This deterministic, step-level feedback acts as a strict grounding mechanism, drastically pruning invalid exploration spaces.
> We summarize these fundamental differences in the table below:
>
> **Table 1. Comparison of FLV with existing formal verification approaches for reasoning.**
> | Method | Primary Goal | Typical domains  | When verification is applied  | Verifier used as training-time |
> |--|--|--|--|--|
> | CLOVER [1] | Reduce execution errors | Logic reasoning | Post translation  | No
> | VANESSA [2] | Benchmark deductive CoT | Deductive reasoning  | Post-hoc | labels/scores for evaluation | No
> | VeriCoT [3] | Validate CoT logic| Biomedical, Legal, Logical reasoning  | Post CoT | No (Offline DPO/SFT only)
> | **FLV (Ours)** | **Improve intrinsic reasoning validity** | **Logical, General, Math**  | **In-loop (during CoT)** |  **Yes (In-loop grounding)**
>
> *References*
>
> [1] Ryu et al., Divide and Translate: Compositional First-Order Logic Translation and Verification for Complex Logical Reasoning. ICLR 2025.
>
> [2] Sadeddine et al., Verifying the Steps of Deductive Reasoning Chains. ACL 2025 Findings.
>
> [3] Feng et al., VeriCoT: Neuro-symbolic Chain-of-Thought Validation via Logical Consistency Checks. arXiv 2025.
>
> > **W2 Cross-Domain Generalization and Additional Baselines.**
>
> **1. Cross-Domain Generalization**
>
> We respectfully clarify that our existing evaluation suite already extends well beyond math and logic. It covers diverse scientific domains via GPQA-D (physics, chemistry, biology) and TheoremQA (physics, EE, CS, finance).
>
> To further demonstrate non-mathematical generalization, we evaluated FLV on LegalBench-SARA (US tax law statutory reasoning). FLV-RL successfully generalizes, outperforming all baselines.
>
> **Table 2. Model performance comparison on LegalBench-SARA (Entailment).**
> | Model | LegalBench-SARA  |
> |--|--|
> |SimpleRL-Zoo | 54.4
> |General-Reasoner | 48.5
> |RLPR | 61.0
> |SynLogic | 60.7
> |ZeroTIR | 48.2
> |SimpleTIR | 57.7
> |**FLV-RL (Ours)** | **64.7**
>
> **2. Comparison with More Tool-Integrated Approaches**
>
> Direct comparison with some tool-integrated models can be challenging due to differences in base models. Our primary goal is to propose an efficient training paradigm rather than engineering a SOTA model.
>
> Nevertheless, to address your concern, we evaluated a recent tool-integrated approach, CoRT-1.5B-RL [1] (based on DeepSeek-R1-Distill-Qwen-1.5B), under our unified protocol. As shown below, FLV demonstrates superior average performance across logical, mathematical, and general domains.
>
> **Table 3: Comparison between CoRT and FLV (Ours) under the unified protocol.**
> | Model | Logical |  | Mathematical |  | General |  | AVG |
> |--|--|--|--|--|--|--|--|
> |  | **KOR-Bench** | **BBH** | **MATH500** | **AIME24** | **GPQA-D** |**TheoremQA** |
> | CoRT-1.5B-RL | 28.0 | 41.0 | 76.8  | 25.2 | 8.6 | 46.5  | 37.7
> | FLV-SFT (Ours) | 48.0 | 68.5 | 77.2 | 20.0 | 32.3 | 53.0 | 49.8
> | **FLV-RL (Ours)** | **51.0** | **70.0** | **78.6** | **20.8** | **35.4** | **55.7** | **51.9**
>
> *Reference*
>
> [1] Li C, Tang Z, Li Z, et al. Cort: Code-integrated reasoning within thinking[J]. arXiv preprint arXiv:2506.09820, 2025.
>
> > **Q2: Training time and computational overhead.**
>
> We acknowledge the overhead of iterative tool execution (e.g., Z3) during RL rollouts. However, this is justified by exceptional sample efficiency: as shown in Table 1 of our submission, FLV-SFT alone outperforms all RL-trained 7B baselines. The high-fidelity feedback yields superior capabilities that offset the per-step time cost. We will add explicit wall-clock times and a cost-performance analysis to the revised version.
>
> ---
> Finally, we express our heartfelt gratitude for your insightful questions and suggestions. They have been invaluable to our work and will be integral to our revised manuscript. We hope our efforts will adequately address your concerns and that you will recognize our work.

---

> > ### Author Rebuttal · Reviewer_DG1B · 2026-04-04
> >
> > I thank the authors for their answers.

---

> > > ### Author Response · Authors · 2026-04-05
> > >
> > > Dear Reviewer DG1B:
> > >
> > > We are glad that our previous responses have adequately addressed your concerns. If you have other questions, we are glad to address them until the end of the rebuttal.

---

### Official Review · Reviewer_33rs · 2026-03-13

**Soundness:** 2
**Presentation:** 2
**Significance:** 3
**Originality:** 2
**Overall Recommendation:** 4
**Confidence:** 3

**Summary:**

The paper proposes using interleaved natural reasoning and formal logic reasoning steps with verification to improve chain-of-thought performance. To train the models, authors first obtain SFT (supervised fine tuning) data from a strong teacher model. In this process, the expected execution output is also obtained from the LLM and compared against the actual execution output. Any potential discrepancies are used to either discard the sample or revise the natural language reasoning trace, in order to improve the natural-formal reasoning alignment. Then, the model is trained further using GRPO. A hierarchical reward design enforces structural and logical correctness in that prioritized order. Experiments using Qwen2.5-7B and Qwen2.5-14B on a variety of different benchmarks.

**Compliance With Llm Reviewing Policy:**

Affirmed.

**Final Justification:**

## Addressed Concerns

1. **Presentation Issues:** The rebuttal and further discussion revealed that the misunderstanding around Section 5.4 was worse than I initially thought. However, the latest reply clarified this. I trust the authors will address these presentation issues in accordance with the feedback. I agree with the authors that the experiment in Section 5.4 is valuable but the presentation must be improved in the revision.
2. **Experiments:** The authors provided numerous additional experimental results across all rebuttals. Since the paper is experiment-driven, and since the most major concerns in my initial review were about this (under Soundness section), the new experiments have strengthened the paper substantially.

## Partially Addressed

1. **Incremental:** As other reviewers also noted, the work is incremental (Reviewer RtUR) and driven by experiments rather than theoretical results (Reviewer spzh). I still think it's a technically solid paper that contributes a meaningful advancement to ML, but I can not confidently say that it will be a very high-impact paper at this point, which brings me to the next point.
2. **Overambitious Claims:** While the incremental nature of the work does not justify a low score on its own, I believe the authors made overambitious claims in the first round of rebuttal, e.g., "a fundamental paradigm shift", "drastically reduces logical hallucinations". Extraordinary claims harm scientific rigor if presented without extraordinary evidence.
   - The authors did soften such claims in their last reply, and provided additional empirical support.
   - But I would recommend softening the core "fundamental paradigm shift" claim in the revision. I don't think this shift is as fundamental as claimed. Because technically, prior methods can also utilize Z3 since it is a Python library. Even though the usage frequency of Z3 increases with this method as the authors noted in their last reply, the sole fact that prior methods utilize Z3 shows it's not a fundamental shift.

## Remaining Concerns

1. **Scalability issues:** Although the authors argue that external LLMs are used solely as data synthesis tools during initial dataset construction, there is a concern that larger models may require more upfront data and external LLMs may become a bottleneck. Unfortunately, we couldn't explore this further since the discussion focused on other matters.

## Final Recommendation

In accordance with the points detailed above, I revise my recommendation from **3 (weak reject)** to **4 (weak accept)**, with my primary reason for shying away from 5 being that I can not confidently predict the impact of this work due to its relatively incremental nature.

**Key Questions For Authors:**

1. Could you please comment on the discrepancy in the SimpleTIR results, discussed in the previous section?
2. Could you please resolve the confusion about Section 5.4, detailed in the previous section?
3. What is the reason for using many different LLMs (DeepSeek-R1, GPT-4o, Claude-Sonnet-4.5) during dataset construction (as detailed in Section 5.1) instead of a single strong model? How did you determine these models?

**Limitations:**

Limitations are described in the appendix in an unlabeled section (page 12). I suggest labeling this section and also discussing the limitations briefly in the main text. Furthermore, I think the current limitations section is incomplete. The following limitations should be discussed in detail:
1. Dependence on a strong teacher model for data generation.
2. Lack of theoretical guarantees.

**Strengths And Weaknesses:**

## Significance

The problem of reconciling formal verification and LLM reasoning is an interesting and relevant problem.

## Originality

Both interleaved reasoning and integration of formal verification to LLMs are widespread in the literature as the authors describe in the paper. Authors claim that combination of these two factors makes their work unique (left column, line 99). However, the pre-existing code-integrated reasoning models such as SimpleTIR and CoRT (both cited in the submission) can already use formal verification tools. The combination of SFT and GRPO is also common in the literature, with the primary novel contribution being the data synthesis pipeline. Moreover, domain-specific nature of other methods are cited as a reason why the proposed method is original (left col., line 102), yet the authors acknowledge in the limitations section (page 12) that the proposed method suffers outside highly structured domains like math and logic. As a result, I think the work is not as original as the authors claim.

## Soundness

Despite the integration of formal reasoning steps, the paper remains an experiment-driven paper since the interleaving of formal and natural reasoning does not inherently remove the source of hallucinations (e.g., probabilistic sampling, natural language). The experimental section evaluates many different models on many different benchmarks, which shows rigor and effort. However, there are some shortcomings:
1. SimpleTIR paper reports 50.5 for AIME24 (vs. 41.0 in submission) and 88.4 for MATH500 (vs. 82.6 in submission). Since this difference is not negligible, I would expect more discussion about this discrepancy in the paper (e.g., is it due to differences in decoding strategy?).
2. CoRT paper seems highly-relevant with code interpreter integration. Although CoRT produces 32B and 1.5B models (unlike 7B and 14B used in this submission), even the 1.5B model outperforms the proposed 7B model on AIME24 (avg@16). CoRT is cited in the submission but neglected in the experiments section.
3. The scalability claims (such as in line 34 in abstract) are not supported sufficiently. The proposed method acts somewhat similarly to a distillation method, requiring a much stronger model to generate the training data. This contrasts with other approaches that rely only on RL. I understand that experimenting with larger models is exceedingly costly. However, the overreliance on a stronger teacher model obviously limits scalability.
4. Extending on the previous point, RL-only variation was not considered in the ablation studies.
5. Ablations with the 14B model would also be valuable.

## Presentation

The paper has a good flow and the main figure (Fig. 3) is well-prepared. However, I found Section 5.4 confusing. It appeared like a new method proposition at the end of the experiments section. I still don’t understand the details of the flexible verification strategy. The referenced appendix section (Appendix E) provides motivation but not the details. Also, it appears that none of the columns in the tables match the Enforced results in Figure 6, and it’s unclear whether the 7B or 14B model was used in Fig. 6.

The texts seem misplaced on the pie chart in Figure 7. Another problem is that the code sections in the appendix are missing indentation.

---

> ### Author Rebuttal · Authors · 2026-03-31
>
> Dear Reviewer 33rs,
>
> Thank you for your assessment and the constructive questions! Below, we provide a comprehensive response to the concerns and questions you raised.
>
> ---
>
> > **Originality & Novelty: SimpleTIR/CoRT already use formal tools.**
>
> We respectfully clarify that our work is a fundamental paradigm shift, not a simple integration. Core novelties:
>
> **1. The "Formal Tool-as-Verifier" Paradigm:** Unlike SimpleTIR/CoRT which use code merely as calculators for narrow math tasks, we use formal solvers (e.g., Z3) to verify the logical soundness of intermediate steps across general domains.
>
> **2. Deterministic Training & Data:** Integrating formal tools into open domains is highly non-trivial. We introduce a novel three-stage validation pipeline for high-fidelity data construction.
>
> **3. Flexible Verification Strategy:** Decoupling calculation from logical validation (§5.4) enables formal verification in non-mathematical tasks where naive tool integration fails.
>
> Achieving SOTA with only ~17k samples proves this is a highly novel paradigm, not mere engineering.
>
>
> > **Soundness 1 & Q1: SimpleTIR result discrepancy (AIME24: 50.5 vs. 41.0; MATH500: 88.4 vs. 82.6).**
>
> This stems from evaluation protocols. SimpleTIR's original scores used larger sampling budgets (**temp=1.0, avg@32, 5 turns**). Our reproduced scores (41.0/82.6) use a strict unified protocol following RLPR [1]: **temp=0.0, avg@16, 4 turns** for AIME24, temp=0.0, 4 turns for MATH500 (Details can be seen in Section 5.1).
>
> [1] Yu T, Ji B, Wang S, et al. Rlpr: Extrapolating rlvr to general domains without verifiers[J]. arXiv preprint arXiv:2506.18254, 2025.
>
>
> > **Soundness 2: CoRT comparison missing.**
>
> While direct comparisons are nuanced due to different base models (CoRT uses DeepSeek-R1-Distill-Qwen-1.5B), we agree this provides valuable context.
>
> Under our unified protocol (max 4 calls, greedy), CoRT scores 25.2 on AIME24 (down from its original 15-call budget). Our FLV models outperform CoRT-1.5B-RL on average.
>
> **Table 2: CoRT vs. FLV (Ours) under the unified protocol.**
> | Model | Logical |  | Mathematical |  | General |  | AVG |
> |--|--|--|--|--|--|--|--|
> |  | **KOR-Bench** | **BBH** | **MATH500** | **AIME24** | **GPQA-D** | **TheoremQA**
> | CoRT-1.5B-RL | 28.0 | 41.0 | 76.8  | **25.2** | 8.6 | 46.5  | 37.7
> | FLV-SFT (Ours) | 48.0 | 68.5 | 77.2 | 20.0 | 32.3 | 53.0 | 49.8
> | FLV-RL (Ours) | **51.0** | **70.0** | **78.6** | 20.8 | **35.4** | **55.7** | **51.9**
>
> > **Soundness 3 & Limitations 1: Scalability / dependency on strong teacher model.**
>
> We would like to clarify that external LLMs are used solely as **data synthesis tools** during initial dataset construction, not as teacher models for knowledge distillation. Subsequent training and inference rely entirely on the target model and the deterministic verifier.
>
> > **Soundness 4: RL-only ablation (no SFT cold start).**
>
> We agree this is an important baseline. Our experiments show that applying RL without SFT initialization (FLV-ZeroRL) fails to teach the model our "Formal Tool-as-Verifier" paradigm, yielding inadequate improvement (**base: 30.3, FLV-ZeroRL: 42.7, FLV-RL: 51.9**). We will include detailed results in the revised version.
>
> > **Soundness 5: 14B ablation.**
>
> Full 14B RL ablations require substantial computational resources beyond our current capacity. However, Table 1 shows 14B step-wise improvements closely mirror 7B trends, demonstrating consistent scaling.
>
> > **Q3: Why multiple teacher models?**
>
> Our work is not traditional knowledge distillation. We use LLMs purely as data synthesis tools to meet formal verification requirements, chosen for efficiency, formatting, and cost. We primarily utilized Claude 4.5 Sonnet due to its superior perfect conversion rate (68.9%), which minimizes discarded data and costs (prompts in Appx I).
>
> **Table 4. Conversion success rate comparision between different models.**
> | Range              | Claude 4.5 Sonnet | DeepSeek-V3.2   | Qwen3-235B-A22B-Instruct     |
> |--------------------|------------------|-----------------|-----------------|
> | 0–99 (discard)             | 23 (22.3%)       | 47 (45.6%)      | 35 (34.0%)      |
> | 99–100 (Stage 2 filtering)     | 9 (8.7%)         | 6 (5.8%)        | 3 (2.9%)        |
> | 100 (accepted)                | 71 (68.9%)       | 50 (48.5%)      | 65 (63.1%)      |
>
>
> > **Presentation Issues**
>
> - Section 5.4: This is an engineering design choice during SFT data construction (decoupling calculation from mandatory verification), not a new method.
>
> - Figure 6: Reports on a held-out 7B training subset, hence the Table 1 mismatch. We will add explicit labels.
>
> - Figure 7 & Appendix: We will fix the pie chart text and code indentation.
>
>
> ---
>
> Finally, we express our heartfelt gratitude for your insightful questions and suggestions. They have been invaluable to our work and will be integral to our revised manuscript. We hope our efforts will adequately address your concerns and that you will recognize our work.

---

> > ### Author Rebuttal · Reviewer_33rs · 2026-04-04
> >
> > I appreciate the rebuttal and new experimental results. Although some of my concerns were addressed, I have follow-up questions:
> >
> > 1. Regarding Section 5.4, you call it "an engineering design choice during SFT data construction". My understanding is that decoupling calculation and mandatory verification requires changes to the inference as well. Could you please explain?
> > 2. Was the method in Section 5.4 used in experiments prior to Section 5.4?
> > 3. I noticed the new LegalBench-SARA experiment (in response to Reviewer DG1B). Thank you for this additional experiment. Was the method in Section 5.4 used in that experiment?
> > 4. "Figure 6: Reports on a held-out 7B training subset, hence the Table 1 mismatch." What's the motivation for using a held-out training dataset for this experiment?
> > 5. Overall, I'm not satisfied with the explanation regarding Section 5.4 (and Fig. 6, which is a part thereof). I totally understand that the authors couldn't clarify this in sufficient detail probably due to character limits. Please use this opportunity to clarify. The authors mention Section 5.4 as a major point in core novelties section in rebuttal (which makes it even more peculiar that it received very limited attention in the main text), yet it seems to me that it makes the method more similar to prior work (relaxing formal verification step and using the coding capabilities for calculation).
> > 6. Extending on the previous point, the authors claim "Unlike SimpleTIR/CoRT which use code merely as calculators for narrow math tasks, we use formal solvers (e.g., Z3)..." I don't think this shift is as fundamental as claimed. Because technically, prior methods can also utilize Z3 since it is a Python library. Just to clarify, I'm not denying that there are some valuable novel aspects in the proposed method. But calling it "a fundamental paradigm shift" at this point does not sound realistic.
> > 7. Regarding the claim that "FLV drastically reduces logical hallucinations versus standard LLMs" (mentioned in reply to Reviewer spzh): I understand that reducing hallucinations is a primary motivation, and the integration of formal verification appears to address this in a straightforward way (due to increased performance in benchmarks). However, as the authors recognize, the method does not provide any end-to-end theoretical guarantees, and the data synthesis pipeline is still dependent on LLMs. Given these, is there any empirical evidence that the hallucinations are actually reduced? Note that improved performance does not necessarily imply reduced hallucinations, as discussed in Fig. 1 (it's possible to reach the correct answer using inconsistent logic).
> >
> > Thank you.

---

> > > ### Author Response · Authors · 2026-04-07
> > >
> > > Dear Reviewer 33rs,
> > >
> > > Thank you for your continued engagement. Your questions highlight areas where our method was not clearly communicated in the text. We deeply appreciate the opportunity to clarify.
> > >
> > > ---
> > > **Response to 1, 2, 3 &4 (Clarifying Section 5.4):**
> > >
> > > We sincerely apologize for the confusion. To clarify: **the "Enforced Verification" discussed in Section 5.4 was a comparitive prototype of the FLV paradigm.** It is not the final method used in our main evaluations.
> > >
> > > In "Enforced" Verification, we naively required every single step to follow a strict "first-reason-then-verify" paradigm. However we found that even in general-domain, pure mathematical calculation takes up a non-negligible portion. Under the "Enforced", the model would first calculate in natural language, then use Python merely to verify—which was highly redundant.
> > >
> > > Realizing that pure calculation doesn't require post-verification (in-time calculation is sufficient and more efficient), we evolved to the **"Flexible" reasoning paradigm**, which intelligently combines in-time calculation with post-verification for logical steps.
> > >
> > > To directly answer your questions:
> > > * **Q1 (Inference changes):** Decoupling calculation and verification does *not* require changes to the inference pipeline. Whether performing in-time calculation or formal post-verification, the underlying mechanism remains the same `natural reasoning → code → tool execution` loop. The shift from "Enforced" to "Flexible" is reflected solely in what the model generates within this loop, not in the inference script itself.
> > > * **Q2, Q3:** The "Enforced" method in Section 5.4 was **not** used in the main experiments (Sections 5.1–5.3) nor in the new LegalBench-SARA experiment. All main experiments evaluate our final, optimized **Flexible** reasoning paradigm.
> > > * **Q4 (Motivation for using a held-out training dataset):** We used a held-out training dataset simply to conduct an additional, supplementary experiment to analyze the model's behavior under different paradigms. Section 5.4 is included purely as an explorative ablation study to justify why we selected the flexible paradigm over a strictly enforced one. We included it because we believe documenting this architectural evolution is meaningful for future research.
> > > ---
> > >
> > > **Q5 & Q6 (Distinction from prior work):**
> > >
> > > We emphasize that FLV is not merely a trivial extension or "relaxation" of prior code-as-calculator methods. It actively integrates two distinct capabilities: in-time calculation and formal logical verification.
> > >
> > > First, while prior code-based methods *could* import Z3, they struggle to invoke it effectively. As shown in Figure 4, baseline models successfully utilize symbolic/logic packages only **42.5%** of the time (compared to FLV-RL: 62.5%). We also manually checked a few cases, finding that baselines like SimpleTIR often fail to call symbolic/logic packages, even when formal verification is strictly necessary. FLV explicitly trains the model on *when* and *how* to formulate these logical constraints, moving beyond using Python merely as a calculator.
> > >
> > > Second, our **FLV-ZeroRL** results confirm that this integration is a newly learned capability rather than a simple elicitation of inherent coding skills. When directly prompted to combine formal verification and calculation without SFT initialization, the model yielded inadequate improvements (**Base: 30.3; FLV-ZeroRL: 42.7; FLV-RL: 51.9**). If our method were simply relaxing existing skills, ZeroRL should have easily adapted. We will include these details in the revised version.
> > >
> > > ---
> > > **Response to 7 (Logical hallucinations)**:
> > >
> > > We agree that improved accuracy does not guarantee reduced hallucinations, and we will soften the claim "drastically reduces logical hallucinations" in the revision.
> > >
> > > Instead, we emphasize that FLV provides a *structural mechanism* against them. In standard LLMs, flawed logic easily propagates. In FLV, intermediate steps are translated into formal constraints (e.g., Z3); a hallucinated step results in an "Unsat" return or error, interrupting the flawed reasoning chain.
> > >
> > > To provide empirical evidence, we analyzed three rollout steps sampled during RL training, utilizing Qwen3-235B-A22B-Instruct to automatically classify the intermediate execution logs. FLV consistently generates logical constraints, and as training progresses, the proportion formally proven correct (`Satisfiable`) steadily increases:
> > >
> > > | RL Rollout Step | Formally Satisfiable (`Sat`) |
> > > | --| -- |
> > > | Step 20 | 14.7%
> > > | Step 40 | 16.9%
> > > | Step 60 | 17.0%
> > >
> > > This trend demonstrates the model learns to generate increasingly sound reasoning over time. We will also add qualitative case studies in the Appendix comparing FLV's solver-grounded traces with standard LLMs to further illustrate this.
> > >
> > > ---
> > > Thank you again for pushing us to clarify these nuances. Your feedback has been incredibly valuable in strengthening the paper!

---

### Official Review · Reviewer_RtUR · 2026-03-14

**Soundness:** 3
**Presentation:** 2
**Significance:** 2
**Originality:** 2
**Overall Recommendation:** 4
**Confidence:** 3

**Summary:**

This paper presents a formal logic-verification-guided framework that interleaves formal symbolic verification with the natural-language generation process.

**Compliance With Llm Reviewing Policy:**

Affirmed.

**Key Questions For Authors:**

1. Can the main result be reproduced using open-source models?

**Limitations:**

yes

**Strengths And Weaknesses:**

### Strengths

1. The SFT stage, execution-based validation pipeline, and verifier-guided RL stage fit together well.

### Weaknesses

1. The novelty in this paper is limited to combining several existing techniques.

2. The training cost seems to be 2 times larger than the baseline.

---

> ### Author Rebuttal · Authors · 2026-03-31
>
> Dear Reviewer RtUR,
>
> Thank you for your positive assessment and the constructive questions! Below, we provide a comprehensive response to the concerns and questions you raised.
>
> ---
>
> > **W1: The novelty in this paper is limited to combining several existing techniques.**
>
> Thank you for proposing this insightful question. We respectfully clarify that our work is a fundamental paradigm shift, not a simple combination. Open-domain reasoning desperately requires rigorous, step-level process verification, which probabilistic model-based verifiers (e.g., LLM-as-a-judge) cannot reliably provide. Our core novelties include:
>
> **1. The "Formal Tool-as-Verifier" Paradigm:** While prior methods (SimpleTIR, CoRT) use code merely as calculators for narrow math tasks, we use formal solvers (e.g., Z3) to verify intermediate logical soundness. This provides deterministic feedback (counterexamples, satisfiability) for active error correction, extending formal rigor to broader general and logical domains. Detailed case studies can be found in Appendix H of our submission.
>
> **2. Deterministic Training & Data:** Integrating formal tools into open domains is highly non-trivial. We introduce a novel three-stage validation pipeline for high-fidelity data construction.
>
> **3. Flexible Verification Strategy:** A naive combination of existing tools fails in open domains due to rigid constraints. Our flexible strategy (§5.4) decouples calculation from logical validation, uniquely enabling formal verification in non-mathematical tasks.
>
> Achieving SOTA performance with only ~17k samples validates that our framework is a highly novel, effective paradigm, not a mere engineering integration.
>
> > **W2: Training cost seems to be 2 times larger than the baseline.**
>
> We acknowledge the increased training time, which is an expected trade-off for our rigorous verification paradigm.
>
> **1. Source of the Overhead:** The baseline operates purely on natural language generation. In contrast, our framework's overhead arises entirely from the iterative execution of the formal verifier (external tool calls) during RL rollouts, which naturally takes more time than generating text alone.
>
> **2. Exceptional Sample Efficiency:** This computational investment is highly justified by the significant performance gains. As shown in Table 1 of our submission, our FLV-SFT alone (without any RL) already outperforms all RL-trained baselines at the 7B scale (avg 49.8 vs. RLPR 47.0). This demonstrates that our high-fidelity data and framework achieve superior results with exceptional sample efficiency, effectively offsetting the per-step time cost.
>
> We will add explicit wall-clock training times and a detailed cost-performance analysis in future versions to make this trade-off transparent.
>
> > **Q1: Reproducibility with open-source models.**
>
> We acknowledge the use of closed-source models (Claude 4.5 Sonnet) for data synthesis. However, our choice was driven purely by efficiency, not necessity, and does not affect reproducibility.
>
> **1. Efficiency in Data Collection:** As shown in Table 1 below, Claude 4.5 Sonnet achieves a higher perfect conversion success rate (68.9%) compared to leading open-source models. Using Claude 4.5 Sonnet simply minimizes discarded data and reduces the time and cost of data collection.
>
> **2. Core Conclusions Unaffected:** Open-source models (e.g., DeepSeek-V3.2, Qwen3-235B-A22B-Instruct-2507) are fully capable of generating the exact same high-quality training data. Because our framework relies on deterministic formal verification to filter data, using open-source models would yield the same final data quality; it would just require a larger initial pool of raw data to account for a slightly higher discard rate.
>
> We will add this empirical comparison to the appendix to guide future open-source reproduction.
>
> **Table 1. Conversion success rate comparision between different models.**
> | Range              | Claude 4.5 Sonnet | DeepSeek-V3.2   | Qwen3-235B-A22B-Instruc      |
> |--------------------|------------------|-----------------|-----------------|
> | 0–99 (discard)             | 23 (22.3%)       | 47 (45.6%)      | 35 (34.0%)
> | 99–100 (Stage 2 filtering)     | 9 (8.7%)         | 6 (5.8%)        | 3 (2.9%)
> | 100 (accepted)                | 71 (68.9%)       | 50 (48.5%)      | 65 (63.1%)
>
> ---
>
> Finally, we express our heartfelt gratitude for your insightful questions and suggestions. They have been invaluable to our work and will be integral to our revised manuscript. We hope our efforts will adequately address your concerns and that you will recognize our work.

---

### Decision · Program_Chairs · 2026-04-30

**Decision:**

Accept (regular)

**Comment:**

The paper proposes FLV, a formal logic verification-guided training framework for LLM reasoning. The method teaches models to interleave natural-language reasoning with formally checkable steps, such as Python/Z3-style verification code and execution feedback. The training pipeline has two stages: first, supervised fine-tuning on synthesized and execution-validated interleaved reasoning traces, and second, GRPO with a hierarchical reward that prioritizes valid formatting, structural compliance, and final-answer correctness. The goal is to move beyond post-hoc checking by using verifier feedback to shape intermediate reasoning behavior.

Reviewers generally found the problem timely and relevant, and they appreciated the broad evaluation across logical, mathematical, and general reasoning benchmarks. The reported gains over natural-language SFT/RL baselines and several reasoning/tool-use baselines are substantial, especially on logic-heavy and theorem-application tasks. The ablations also support the importance of formal verification guidance: FLV-SFT substantially outperforms Natural-SFT, and FLV-RL improves further, whereas natural-language RL provides only limited gains. The rebuttal added useful clarifications and experiments, including comparisons under a unified protocol, an RL-without-SFT ablation, open-source data-synthesis feasibility checks, additional LegalBench-SARA results, and more detailed synthesis-pipeline statistics.

The main weaknesses concern novelty, positioning, cost, and presentation. Several reviewers noted that tool-integrated reasoning and formal verification for LLM reasoning are already active areas, and that the contribution is closer to a strong system and training-pipeline integration than a fundamental new paradigm. In particular, prior code-integrated approaches can in principle invoke formal tools such as Z3, so the authors should avoid overstating the distinction as a "fundamental paradigm shift." The method also depends on a nontrivial data synthesis pipeline using strong external models and incurs additional tool-execution overhead during training.

After considering the reviews and rebuttal, I find the positive evidence sufficient for acceptance. The remaining concerns are real but mostly affect framing and scope rather than the basic validity of the empirical contribution. The paper demonstrates that training models to use formal tools as intermediate verifiers, rather than only as calculators or post-hoc filters, can improve reasoning across a reasonably broad set of benchmarks. Therefore, I recommend acceptance.